# Metric-Normalized Posterior Leakage (mPL): Attacker-Aligned Privacy for Joint Consumption

## Abstract

*Metric differential privacy (mDP)* strengthens *local differential privacy (LDP)* by scaling noise to semantic distance, but many machine learning (ML) systems are consumed under joint observation, where model-agnostic, per-record guarantees can miss leakage from evidence aggregation. We introduce *metric-normalized posterior leakage (mPL)*, an attacker-aligned, distance-calibrated measure of posterior-odds shift induced by releases, and show that for single or independent releases, uniformly bounding mPL is equivalent to mDP. Under joint observation, however, satisfying mDP may still leave mPL high because learned aggregators compound evidence across correlated items. To make control practical, we formalize *probabilistically bounded mPL (PBmPL)*, which limits how often mPL may exceed a target budget, and we operationalize it via *Adaptive mPL (AmPL)*, a trust-and-verify framework that perturbs, audits with a learned attacker, and adapts parameters (with optional Bayesian remapping) to balance privacy and utility. In a word-embedding case study, neural adversaries violate mPL under joint consumption despite per-record mDP perturbations, whereas AmPL substantially lowers the frequency of such violations with low utility loss, indicating PBmPL as a practical, certifiable protection for joint-consumption settings.

## 1. Introduction

*Local differential privacy (LDP)* (Duchi et al., 2013) enforces a uniform indistinguishability requirement between any two inputs, regardless of how similar those inputs are. This metric-agnostic stance is often a poor fit for modern continuous domains and embedding spaces (Imola et al.,

[1]Anonymous Institution, Anonymous City, Anonymous Region, Anonymous Country. Correspondence to: Anonymous Author <anon.email@domain.com>.

Preliminary work. Under review by the International Conference on Machine Learning (ICML). Do not distribute.

2022), where distances encode semantics: nearby points are naturally similar, while far-apart points may correspond to qualitatively different meanings. As a result, uniform protection can lead to an unfavorable utility–privacy trade-off, either injecting excessive noise or failing to adequately protect fine-grained neighborhoods.

*Metric-aware* privacy notions address this mismatch by incorporating geometry into the guarantee. In particular, *metric differential privacy (mDP)* (Chatzikokolakis et al., 2013) (also called *Lipschitz privacy* (Koufogiannis et al., 2015)) requires that the indistinguishability between secrets degrades smoothly with their metric distance, a formulation that is especially natural for locations and embeddings. In the location-privacy literature, *geo-indistinguishability* (Andrés et al., 2013) popularized practical mechanisms (e.g., planar Laplace noise) and motivated optimization-based obfuscation tailored to road networks, points of interest, and mobility priors. Subsequent work (Bordenabe et al., 2014; Liu & Qiu, 2025) studied optimal mechanism design under metric constraints, related utility to transport/Wasserstein-style costs, and developed composition and group-privacy analyses in metric spaces. Complementing these defenses, recent studies have analyzed inference risks under correlated releases and proposed context-aware perturbation methods for metric-aware guarantees (Qiu et al., 2022; Yadav et al., 2024).

Although mDP incorporates geometry, it is typically formulated as a bound on output-distribution ratios between pairs of inputs and is often analyzed on a per-release basis. In contrast, practical adversaries reason about *posterior beliefs* and routinely aggregate *correlated* observations, for example, multiple perturbed locations along a trajectory (Yadav et al., 2024) or multiple perturbed tokens within a sentence (Staab et al., 2024). These settings suggest that per-release guarantees alone may not faithfully reflect inferential risk under joint consumption. Accordingly, we shift the evaluation target from isolated output ratios to the extent to which an attacker's beliefs *sharpen* after jointly observing multiple, potentially dependent releases, particularly in the presence of learned inference models that can effectively combine such evidence.

## Our Contributions

**(1) From per-release ratio bounds to metric-normalized inferential leakage.** We rethink mDP for *joint-consumption* settings by introducing *metric-normalized posterior leakage (mPL)*, a geometry-aware, attacker-aligned criterion that quantifies the metric-calibrated shift in posterior odds between candidate secrets after observing releases (*Definition 2.2*). We define *bounded mPL* by requiring mPL to lie within a budget $\epsilon$ (*Definition 2.3*) and establish *post-processing invariance* (*Proposition 2.4*). Importantly, for a single release and independent compositions, we prove that uniformly bounding mPL is equivalent to $\epsilon$-mDP (*Propositions 2.5–2.6*), showing that mPL *recovers* mDP in the regime where per-release analysis applies.

**(2) Joint-consumption leakage under dependence with learned evidence aggregation.** *However, the connection between mDP and mPL breaks under dependent joint consumption*, i.e., even when a mechanism satisfies *per-record* mDP, an attacker can still achieve substantial belief sharpening (large mPL) by aggregating correlated releases (sets/sequences). We demonstrate this gap both with an explicit joint secret model and with model-free learned aggregators, instantiating the attacker with an RNN, an LSTM, and a Transformer (Vaswani et al., 2017), which reveal non-trivial mPL violations for standard mDP mechanisms such as the exponential mechanism (Chatzikokolakis et al., 2015). This is related in spirit to prior *posterior-based* privacy frameworks (e.g., *Pufferfish* (Kifer & Machanavajjhala, 2012)), but these works are typically studied in the *central* setting and rely on an *explicit* class of data-generating distributions, whereas mPL is *local*, *metric-normalized*, and designed to be *audited* under implicit dependencies.

**(3) Auditable tail-risk control and attacker-in-the-loop calibration.** To move beyond worst-case control under dependence, we introduce *probabilistically bounded mPL (PBmPL)* (*Definition 3.1*), which bounds the *probability* that mPL exceeds a prescribed privacy budget, together with a sampling-based auditing procedure with confidence guarantees. Building on this audit, we develop *Adaptive mPL (AmPL)*, a *trust-and-verify* framework that (i) applies level-wise perturbation, (ii) audits mPL/PBmPL using learned posterior estimators trained on joint observations, and (iii) adaptively updates perturbation strengths using attacker feedback to balance privacy and utility. Optionally, AmPL applies *Bayesian remapping* (Chatzikokolakis et al., 2017) as post-processing to recover utility without weakening privacy, by post-processing invariance (*Proposition 2.4*).

**(4) Case study: joint-consumption privacy for text embeddings.** We instantiate our framework on text embeddings, perturbing *personally identifiable information (PII)* and *potentially identifying information (PoII)* under joint consumption. Empirically, we find that standard per-record mDP mechanisms (e.g., the exponential mechanism) can still incur substantial mPL violations under neural aggregation. For example, a Transformer-based inference attacker yields mPL $\approx 0.33$ despite the mechanism satisfying mDP. In contrast, AmPL reduces leakage to $\approx 0.12$ while maintaining comparable utility, demonstrating that attacker-in-the-loop calibration can effectively control posterior leakage and revealing the privacy–utility trade-off in a practical embedding setting.

## 2. Metric-Normalized Posterior Leakage

We first introduce the setting and mDP (§2.1) and define mPL and characterize its properties, including equivalence to mDP for single/independent releases (§2.2, §2.3). We then discuss joint consumption with correlated secrets, where per-record mDP may fail to control mPL (§2.4).

### 2.1. Preliminaries

We study *local perturbation* mechanisms that randomize each record before release. Formally, a mechanism $\mathcal{M}$ is a randomized mapping $\mathcal{M} : \mathcal{X} \to \mathcal{Y}$, where $\mathcal{X}$ is the domain of secret records and $\mathcal{Y}$ is the domain of released (perturbed) records. The semantic similarity between secrets is captured by a distance function $d : \mathcal{X} \times \mathcal{X} \to \mathbb{R}_{\geq 0}$; we write $d_{x_i,x_j}$ for the distance between any $x_i, x_j \in \mathcal{X}$. We consider *joint consumption* of $L$ secret records, represented as $\mathbf{x} = (x^1, \ldots, x^L) \in \mathcal{X}^L$. For each $\ell \in [L]$, let $X^\ell$ and $Y^\ell$ denote the random variables corresponding to the $\ell$-th secret record and its released output, respectively.

**Definition 2.1** (mDP). Let $(\mathcal{X}, d)$ be a metric secret space and let $\mathcal{M}$ be a perturbation mechanism with input space $\mathcal{X}$ and output space $\mathcal{Y}$. We say that $\mathcal{M}$ satisfies $(\epsilon, d)$-*mDP* if,

$$\sup_{x_i \neq x_j} \sup_{y \in \mathcal{Y}} \ln \frac{\Pr[\mathcal{M}(x_i) = y]}{\Pr[\mathcal{M}(x_j) = y]} \leq \epsilon d_{x_i,x_j}. \tag{1}$$

where $\epsilon$ denotes the *privacy budget*.

Intuitively, mDP requires that small changes in input $x$ induce only bounded changes in the law of $\mathcal{M}(x)$, yielding privacy calibrated to the metric $d$. A smaller $\epsilon$ implies a tighter bound, and hence stronger privacy, so that less can be inferred about $x$ from observing $\mathcal{M}(x)$.

Following (Wang et al., 2017; Liu & Qiu, 2025; Imola et al., 2022), we consider a discrete perturbation space $\mathcal{Y} = \{y_1, \ldots, y_K\}$. To facilitate analysis, we represent $\mathcal{M}$ as a deterministic function $\tilde{\mathcal{M}}$ defined by $\mathcal{M}(x) \equiv \tilde{\mathcal{M}}(x, Z)$, where $Z \sim \text{Uniform}(0,1)$ is an auxiliary random variable that captures the randomness of $\mathcal{M}$. Specifically, we define cumulative sums

$$F_0(x) = 0, \ F_k(x) = \sum_{v=1}^{k} \Pr[\mathcal{M}(x) = y_v], \ k = 1, \ldots, K. \tag{2}$$

Then $\tilde{\mathcal{M}}$ is given by

$$\tilde{\mathcal{M}}(x, Z) = \sum_{k=1}^{K} y_k \mathbf{1}_{[F_{k-1}(x), F_k(x))}(Z), \tag{3}$$

where $\mathbf{1}_{[a,b)}(Z)$ is the indicator function, equal to 1 if $Z \in [a, b)$ and 0 otherwise.

In the following, we use both notations $\mathcal{M}$ and $\tilde{\mathcal{M}}$: $\mathcal{M}$ for simplicity of exposition, and $\tilde{\mathcal{M}}$ in formal arguments (e.g., the proof of Proposition 2.6).

**Threat model.** We adopt standard assumptions (Liu & Qiu, 2025): the server is *honest-but-curious*, follows the protocol, and attempts inference. The attacker is prior-informed, knows the mechanism $\mathcal{M}$ (and its parameters), has auxiliary data from the same population to estimate priors/train a posterior estimator, and can passively aggregate one or more correlated noisy releases to infer $x_\ell$ from a candidate set $\mathcal{X}_\ell$. Given at least a single perturbed record $y$ and the mechanism $\mathcal{M}$, the server can infer the posterior distribution of $X$ using Bayes' rule (Yu et al., 2017),

$$\Pr(X = x \mid \mathcal{M}(X) = y) \tag{4}$$

$$= \frac{\Pr(\mathcal{M}(X) = y \mid X = x) \Pr(X = x)}{\sum_{x' \in \mathcal{X}} \Pr(\mathcal{M}(X) = y \mid X = x') \Pr(X = x')}. \tag{5}$$

This adversarial model reflects realistic deployments in which the server is run by an organization with incentives to collect and analyze user data and thus cannot be fully trusted from a privacy perspective. In contrast, users are assumed to be honest and to faithfully apply the prescribed perturbation protocol before transmitting their data. We do not consider collusion among users, or between users and the server, since such settings are typically outside the standard scope of LDP/mDP-style guarantees and require different threat models and defenses (To et al., 2017).

**Joint observation.** In this paper, we consider attackers who observe *multiple* perturbed releases. Given a user's secret sequence $\mathbf{x} = (x^{(1)}, \dots, x^{(L)})$, a randomized mechanism $\mathcal{M}$ is applied *independently* to each component, producing $\mathbf{y} = (y^{(1)}, \dots, y^{(L)})$ with $y^{(\ell)} = \mathcal{M}(x^{(\ell)})$ for $\ell \in [L]$. The adversary observes the joint output $\mathbf{y}$ and aims to infer the original secrets $\mathbf{x}$. Importantly, while the perturbations are applied independently across $\ell$, the secrets $\{x^{(\ell)}\}_{\ell=1}^{L}$ may be statistically dependent. Then, the adversary estimates the posterior distribution of each secret record $X_\ell$ conditioned on the full perturbed sequence $\mathbf{y}$ as:

$$\Pr[X_\ell = x \mid \{\mathcal{M}(X_1), \dots, \mathcal{M}(X_L)\} = \mathbf{y}], \quad x \in \mathcal{X}. \tag{6}$$

### 2.2. mPL and Its Post-Processing Property

To quantify how much the joint observation $\mathbf{y}$ reveals about the original input $\mathbf{x}$, we define *mPL* as the change in relative likelihood (posterior odds) between two candidate records $x_i$ and $x_j$ from prior to posterior after observing $\mathbf{y}$.

**Definition 2.2** (mPL). *mPL* between a pair of records $x_i, x_j \in \mathcal{X}$ given the joint observation $\mathbf{y} = (y^1, \dots, y^L)$ is defined as

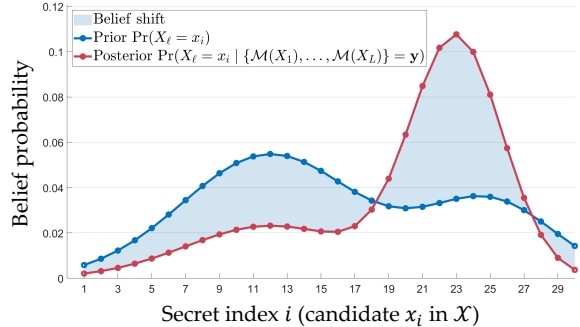

*Figure 1.* Attacker belief update: prior vs. posterior distribution.

$$\mathrm{mPL}_{\mathcal{M}}(x_i, x_j, \mathbf{y}) \tag{7}$$

$$= \frac{1}{d_{x_i, x_j}} \left| \ln \frac{\Pr(X_\ell = x_i \mid \{\mathcal{M}(X_1), \dots, \mathcal{M}(X_L)\} = \mathbf{y})}{\Pr(X_\ell = x_j \mid \{\mathcal{M}(X_1), \dots, \mathcal{M}(X_L)\} = \mathbf{y})} \right.$$

$$\left. - \ln \frac{\Pr(X_\ell = x_i)}{\Pr(X_\ell = x_j)} \right|. \tag{8}$$

Here, the prior ratio $\frac{\Pr(X_\ell = x_i)}{\Pr(X_\ell = x_j)}$ reflects the likelihood of $X_\ell$ being $x_i$ versus $x_j$ before any observation, while the posterior ratio $\frac{\Pr(X_\ell = x_i \mid \{\mathcal{M}(X_1), \dots, \mathcal{M}(X_L)\} = y)}{\Pr(X_\ell = x_j \mid \{\mathcal{M}(X_1), \dots, \mathcal{M}(X_L)\} = y)}$ captures the updated belief after observing $\mathbf{y}$. Fig. 1 visualizes this belief update: the attacker starts with a prior distribution $\Pr(X_\ell = x_i)$ over candidate secrets, and after observing the released outputs $\mathbf{y} = \{\mathcal{M}(X_1), \dots, \mathcal{M}(X_L)\}$, updates to a posterior distribution $\Pr(X_\ell = x_i \mid \mathbf{y})$. The shift from the prior curve to the posterior curve (shaded region) illustrates how the observation concentrates probability mass on some candidates and reduces it on others. *The posterior leakage* $\mathrm{mPL}_{\mathcal{M}}(x_i, x_j, \mathbf{y})$ *thus measures the change in these relative beliefs, normalized by the record distance* $d_{x_i, x_j}$. A smaller leakage value indicates that the perturbation mechanism $\mathcal{M}$ reveals less information, thereby offering stronger privacy protection.

Notably, the key distinction between our posterior inference model (Eq. (6)) and the posterior-based formulation in (Kifer & Machanavajjhala, 2012) (Eq. (4)) is the threat model and how dependencies are handled: our attacker performs *local*, *metric-normalized* inference for each $X_\ell$ under *implicit* correlations learned from data, whereas (Kifer & Machanavajjhala, 2012) is typically framed in the *central* setting and specifies privacy with respect to an *explicit* class of data-generating distributions.

**Definition 2.3** ($\epsilon$-Bounded mPL). A randomized perturbation mechanism $\mathcal{M}$ is said to satisfy $\epsilon$-bounded mPL if, for every pair of distinct secrets $x_i \neq x_j$ and every joint observation $\mathbf{y} \in \mathcal{Y}^L$,

$$\sup_{x_i \neq x_j} \sup_{\mathbf{y} \in \mathcal{Y}^L} \mathrm{mPL}_{\mathcal{M}}(x_i, x_j, \mathbf{y}) \leq \epsilon. \tag{9}$$

**Proposition 2.4** (Post-processing for bounded mPL). *Let* $\mathcal{M} : \mathcal{X} \to \mathcal{Y}$ *be a randomized mechanism that satisfies the*

$\epsilon$-*bounded mPL constraint.* *For any (possibly randomized) function* $f : \mathcal{Y} \to \mathcal{Z}$, *define the post-processed mechanism* $(f \circ \mathcal{M})(x) \triangleq f(\mathcal{M}(x))$, $\forall x \in \mathcal{X}$, *where* $\mathcal{Z} = \mathrm{Range}(f \circ \mathcal{M})$. *Then* $f \circ \mathcal{M}$ *also satisfies the bounded joint mPL constraint:*

$$\sup_{x_i \neq x_j} \sup_{\mathbf{z} \in \mathcal{Z}^L} \mathrm{mPL}_{f \circ \mathcal{M}}(x_i, x_j, \mathbf{z}) \leq \epsilon. \quad (10)$$

*Detailed proof can be found in Appendix D.1.*

Intuitively, Proposition 2.4 implies that, if for every possible perturbed records $\mathbf{y}$, their joint mPL is within the privacy budget $\epsilon$, then post-processing the output using $f(\mathbf{y})$ cannot amplify this ratio, since post-processing coarsens the output space, mixing outcomes, which cannot increase the distinction between $X_\ell = x_i$ and $X_\ell = x_j$.

### 2.3. Properties Based on Individual or Independent Observations

**Proposition 2.5** (Single-observation equivalence of mPL and mDP)**.** *Define the single-observation mPL for a pair* $(x_i, x_j)$ *and observation* $y$ *by*

$$\mathrm{mPL}_{\mathcal{M}}((x_i, x_j), y) \quad (11)$$
$$= \frac{1}{d_{x_i, x_j}} \left| \ln \frac{\Pr(X = x_i | \mathcal{M}(X) = y)}{\Pr(X = x_j | \mathcal{M}(X) = y)} - \ln \frac{\Pr(X = x_i)}{\Pr(X = x_j)} \right|.$$

*For any* $\epsilon \geq 0$, $\mathcal{M}$ *satisfies* $(\epsilon, d)$-*mDP if and only if the single-observation mPL bound holds, i.e.,*

$$\sup_{x_i \neq x_j} \sup_{y \in \mathcal{Y}} \mathrm{mPL}_{\mathcal{M}}((x_i, x_j), y) \leq \epsilon. \quad (12)$$

*A detailed proof appears in Appendix D.2.*

While real-world record often contains dependencies (e.g., between a person's name and organization), analyzing posterior leakage under the simplifying assumption that protected records $X_1, \ldots, X_L$ are *independently distributed* offers useful theoretical insights. Under this assumption, we establish a connection between individual and joint posterior leakage in *Proposition 2.6*:

**Proposition 2.6** (Independent-observation equivalence of mPL and mDP)**.** *If the L secret words* $X_1, \ldots, X_L$ *are independently distributed, then ensuring*

$$\sup_{x_i \neq x_j} \sup_{y^\ell \in \mathcal{Y}} \mathrm{mPL}_{\mathcal{M}}((x_i, x_j), y^\ell) \leq \epsilon \quad (13)$$

*for each* $y^\ell$ ($\ell = 1, ..., L$) *is sufficient to guarantee*

$$\sup_{x_i \neq x_j} \sup_{\mathbf{y} \in \mathcal{Y}^L} \mathrm{mPL}_{\mathcal{M}}(x_i, x_j, \mathbf{y}) \leq \epsilon. \quad (14)$$

*A detailed proof appears in Appendix D.3.*

The proposition shows that without inter-token dependencies, individual-level mDP bounds suffice to ensure privacy under joint observation. However, this assumption rarely holds in practice.

### 2.4. Threat Models based on Joint and Correlated Observations

In this part, we relax the independence assumption and introduce more realistic threat models where the records $X_1, \ldots, X_L$ are dependent.

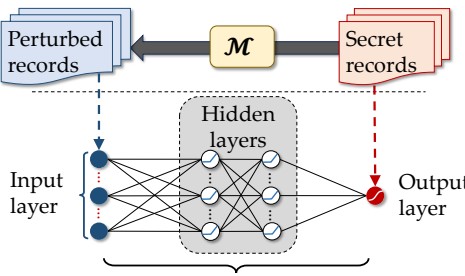

DNN to establish an empirical relationship $f(\cdot)$

*Figure 2.* Threat model.

**(1) Explicit joint-probability attacker (a toy example).** We consider an attacker that models the *joint* distribution of two secrets and performs *Bayesian inference* over two perturbed outputs. Let $X_1, X_2 \in \mathcal{X} = \{x_1, x_2\}$ with a correlated prior:

$$\Pr(X_1 = x_1, X_2 = x_1) = \Pr(X_1 = x_2, X_2 = x_2) = 0.01$$
$$\Pr(X_1 = x_1, X_2 = x_2) = \Pr(X_1 = x_2, X_2 = x_1) = 0.49,$$

and set $\epsilon = 1.0$. For an *exponential mechanism (EM)* perturbation $\mathcal{M}_{\mathrm{EM}}$ with two outputs $\{y_1, y_2\}$, suppose $\Pr(\mathcal{M}_{\mathrm{EM}}(X_i) = y_1 \mid X_i = x_1) = 0.72$, $\Pr(\mathcal{M}_{\mathrm{EM}}(X_i) = y_2 \mid X_i = x_1) = 0.28$, $\Pr(\mathcal{M}_{\mathrm{EM}}(X_i) = y_1 \mid X_i = x_2) = 0.28$, $\Pr(\mathcal{M}_{\mathrm{EM}}(X_i) = y_2 \mid X_i = x_2) = 0.72$. A direct calculation shows that for each $y_k \in \{y_1, y_2\}$,

$$\mathrm{mPL}_{\mathcal{M}_{\mathrm{EM}}}(x_1, x_2, y_k) = 0.944 < \epsilon, \quad (15)$$

so observing each perturbed record *individually* does not violate the mPL bound (therefore also achieving mDP according to Proposition 3.2).

In contrast, when the two outputs are consumed *jointly*, we obtain

$$\mathrm{mPL}_{\mathcal{M}_{\mathrm{EM}}}\big(x_1, x_2, \{\mathcal{M}_{\mathrm{EM}}(x_1), \mathcal{M}_{\mathrm{EM}}(x_2)\} \quad (16)$$
$$= \{y_1, y_2\}\big) = 1.846 > \epsilon, \quad (17)$$

demonstrating that joint consumption under a correlated prior can trigger posterior-leakage violations even when all single-observation checks pass. Full details appear in Appendix C.1.

**(2) Inference models based on implicit joint probability.** Explicit Bayesian joint inference can, in principle, reveal joint leakage, but it is often impractical: computing the exact posterior $p(\mathbf{x} \mid \mathbf{y})$ requires summing (or integrating) over all $\mathbf{x} \in \mathcal{X}^L$, yielding a normalizing constant of size $|\mathcal{X}|^L$ in the discrete case (i.e., exponential in sequence length). Moreover, the joint prior $p(\mathbf{x})$ is typically unknown and any assumed probabilistic model may be misspecified (Bernardo & Smith, 1994).

In these models, we instantiate the attacker as a high-capacity neural posterior estimator that directly approximates $\Pr(X_\ell = x_i | \{\mathcal{M}(X_1), \ldots, \mathcal{M}(X_L)\} = \mathbf{y})$. Specifically, we employ three DNN architectures, *RNN*, *LSTM*,

and *Transformer*, to reconstruct secret records from their perturbed counterparts. As illustrated in Fig. 2, we apply the perturbation mechanism $\mathcal{M}$ to all the secret records and generate corresponding perturbed records. Then we randomly select 80% of secret–perturbed pairs, 60% serving as samples for training and 20% for validation, with each model minimizing the *mean squared error (MSE)* between the predicted and true records. We use the Adam optimizer (Kingma & Ba, 2017) with an initial learning rate of 0.001, reducing it when validation performance plateaus.

*Posterior and prior approximation.* Given an adversary reconstruction $\hat{x} \in \mathcal{X}$, we compute squared Euclidean distances $d^2_{\hat{x}, x_i} = \|\hat{x} - x_i\|^2_2$ to each candidate $x_i \in \mathcal{X}$ and map them to a temperature-scaled Gaussian softmax (Guo et al., 2017):

$$\Pr(X = x_i \mid Y = y) = \frac{\exp\left(-d^2_{\hat{x}, x_i}/(\tau_{\text{base}}\tau)\right)}{\sum_{x_j \in \mathcal{X}} \exp\left(-d^2_{\hat{x}, x_j}/(\tau_{\text{base}}\tau)\right)}, \tag{18}$$

where $\tau_{\text{base}}$ and $\tau > 0$ control the sharpness. We treat this distribution as a numerically stable approximation of the attacker's posterior over $X$ given $y$. If sensitive tokens are deterministically replaced by a fixed placeholder $y_{\text{mask}}$ (e.g., ``xxxx'') rather than perturbed by $\mathcal{M}$, then $y$ carries no information about the secret; the posterior therefore equals the prior.

*Initial results.* According to the case study in Section 4 (PII Protection), learned adversaries reveal non-trivial mPL violations under per-record mDP perturbation mechanism (example distributions in Fig. 4; full results in Table 1).

**Discussion: Per-user accounting.** Per-user budgeting is effective when reliable user identifiers enable per-user accounting; we instead target settings without such identifiers (e.g., many text/embedding datasets) and adopt a user-agnostic formulation that controls joint leakage over arbitrarily correlated secrets (detailed discusion can be found in Appendix C.2).

## 3. Data Perturbation Framework

As discussed in Section 2.4, exact closed-form calibration of mechanism parameters for mPL is generally intractable, since evaluating mPL requires posteriors induced by high-dimensional, correlation-aware likelihoods. To operationalize mPL, we adopt a *trust–and–verify* framework with an attacker in the loop, called *Adaptive mPL (AmPL)*. AmPL starts from a principled per-record perturbation (motivated by the independent case), trains a high-capacity adversary to estimate posteriors from the resulting releases, and then *verifies* (and updates) the mechanism by auditing the resulting mPL estimates. Iterating this loop resolves the chicken-and-egg dependency: mechanism parameters require attacker feedback, while the attacker requires mechanism-perturbed data for training. Finally, AmPL supports *level-wise* protection by stratifying secrets into multiple sensitivity tiers.

Figure 3 illustrates the AmPL framework via a text-embedding example with two sensitivity tiers: *personally identifiable information (PII)* and *potentially identifying information (PoII)*. Here, PII includes attributes that directly identify or authenticate an individual (e.g., full name, email address, phone number, or precise home address). PoII refers to attributes that may not uniquely identify a person in isolation but can materially reduce the anonymity set or reveal sensitive traits when combined with other data (e.g., employer, city of residence, demographic descriptors, or fine-grained preferences). This tiered model enables stricter privacy parameters for PII while still monitoring joint-consumption leakage for PoII.

As the figure shows, in each round, AmPL ① performs **level-wise data perturbation** by partitioning the secret record space $\mathcal{X}$ into $N$ tiers and applying $N$ corresponding perturbation levels to the original representation $\mathbf{x}$, producing a perturbed output $\mathbf{y}$; ② trains an **adversarial DNN** (e.g., RNN/LSTM/Transformer) to reconstruct $\mathbf{x}$ from $\mathbf{y}$ or infer protected attributes, yielding $\hat{\mathbf{x}}$, and uses this inference to evaluate the **mPL violation ratio** as the privacy risk under joint consumption; ③ uses this risk estimate for **feedback-driven adjustment**, iteratively updating perturbation strengths to balance privacy and utility and move toward a target leakage threshold; and ④ applies **Bayesian remapping** $f(\cdot)$ to $\mathbf{y}$ to improve downstream utility, which preserves the privacy guarantee as pure post-processing (Proposition 2.4).

Next, we introduce the details of Steps ①–④.

**Step ①: Level-wise data perturbation.** Let $N \in \mathbb{N}$ denote the number of sensitivity tiers. We first partition $\mathcal{X}$ into disjoint subsets $\{\mathcal{X}^{(1)}, \ldots, \mathcal{X}^{(N)}\}$ and define a level-assignment function $g : \mathcal{X} \to \{1, \ldots, N\}$ that maps each secret $x \in \mathcal{X}$ to its sensitivity level. For each level $\ell \in \{1, \ldots, N\}$, specify a mechanism $\mathcal{M}_\ell(\cdot; \alpha_\ell)$ with privacy/perturbation parameter $\alpha_\ell$. Given $x$, the released output is $y \sim \mathcal{M}_{g(x)}(x; \alpha_{g(x)})$, so that protection strength matches the sensitivity of $x$. The collection $\{\alpha_\ell\}_{\ell=1}^L$ can be tuned (i.e., via feedback control in Step ③) to meet target leakage–utility trade-offs.

*Two-level example for word-embedding privacy (PII vs. PoII):* In this case, we let $N = 2$ with $\mathcal{X}^{(1)}$ denoting direct identifiers (PII) and $\mathcal{X}^{(2)}$ denoting quasi-identifiers (PoII). We assign a stronger perturbation to PII and a milder one to PoII (e.g., mechanisms $\mathcal{M}_1, \mathcal{M}_2$ with $\epsilon_1 < \epsilon_2$), reflecting their different disclosure risks. A detailed design and evaluation of this two-level word-embedding perturbation appear in Section 4 (Case Study) and Appendix E.

**Step ②: Learned Adversary**. To evaluate and mitigate posterior leakage under realistic adversarial settings, we adopt a learned adversary approach based on DNNs, as introduced in Section 2.4. Specifically, models such as RNNs,

*Figure 3.* Illustration of the AmPL Framework (example: protecting PII and PoII word embeddings).

LSTMs, and Transformers are trained to reconstruct the original records from their perturbed versions, effectively simulating strong inference attacks that exploit semantic dependencies across tokens. We then use the outputs of these adversarial models, i.e., the approximated posterior distributions over sensitive tokens, to assess whether the posterior leakage bounds are satisfied.

Notably, enforcing the posterior leakage constraint for *all* secret record pairs and perturbed records can lead to an overly conservative privacy budget. To address this, we adopt a probabilistic relaxation that requires the constraint to hold with high probability rather than deterministically.

**Definition 3.1** (Probabilistic bounded mPL). Given a perturbation mechanism $\mathcal{M}_\ell$ ($\ell = 1, ..., L$), we define the violation probability $p_{\mathcal{X}_\ell^2}$ as the probability that the posterior leakage exceeds the privacy budget $\epsilon$ for a randomly sampled pair of records and output:

$$p_{\mathcal{X}_\ell^2} = \Pr\left[\mathrm{mPL}_{\mathcal{M}_\ell}(X_i, X_j, Y) > \epsilon\right], \quad (19)$$

where $X_i$ and $X_j$ are drawn from the secret record domain $\mathcal{X}_\ell$. We say $\mathcal{M}_\ell$ can achieve $(\delta, \epsilon_\ell)$-PBmPL if $p_{\mathcal{X}_\ell^2} \le \delta$.

Directly computing $p_{\mathcal{X}_\ell^2}$ is computationally prohibitive, as it requires evaluating all $|\mathcal{X}_\ell|^2$ record pairs. In practice, where $|\mathcal{X}_\ell|$ may range from tens of thousands to over one hundred thousand records (e.g., 5,448 PIIs and 5,492 PoIIs in AG-News dataset (Zhang et al., 2015)), exhaustive evaluation is intractable. To address this, we estimate $p_{\mathcal{X}_\ell^2}$ via *random sampling*. Specifically, we uniformly sample a subset $\mathcal{S}_\ell \subseteq \mathcal{X}_\ell^2 \times \mathcal{Y}$ consisting of $S_\ell$ triplets $(x_i, x_j, y)$, and define the empirical estimate:

$$\hat{p}_{\mathcal{S}_\ell} = \frac{1}{S_\ell} \sum_{(x_i, x_j, y) \in \mathcal{S}_\ell} \mathbf{1}\left(\mathrm{mPL}_{\mathcal{M}_\ell}(x_i, x_j, y) > \epsilon\right), \quad (20)$$

where $\mathbf{1}(\cdot)$ denotes the indicator function. Because $\mathcal{S}_\ell$ is sampled uniformly, $\hat{p}_{\mathcal{S}_\ell}$ is an *unbiased estimator* of $p_{\mathcal{X}_\ell^2}$, i.e., $\mathbb{E}[\hat{p}_{\mathcal{S}_\ell}] = p_{\mathcal{X}_\ell^2}$. We further establish the following concentration guarantee:

**Proposition 3.2.** *[Concentration Guarantee for Probabilistic mDP Sampling] If the empirical violation rate satisfies*

$\hat{p}_{\mathcal{S}_\ell} = \xi\delta$ *for some constant* $\xi < 1$, *then* $\Pr\left[p_{\mathcal{X}_\ell^2} \le \delta\right] \ge 1 - 2\exp(-2S_\ell(1-\xi)^2\delta^2)$. *The detailed proof can be found in* **Appendix D.4**.

Proposition 3.2 shows that as the sample size $S_\ell$ increases, the bound $2\exp(-2S_\ell(1-\xi)^2\delta^2)$ rapidly approaches zero, ensuring that $\Pr[p_{\mathcal{X}_\ell^2} \le \delta]$ approaches one.

**Proposition 3.3** (Asymptotic faithfulness of the mPL audit). *[Asymptotic faithfulness of the mPL audit] Let* $p(x \mid y)$ *denote the true posterior and* $q_\theta(x \mid y)$ *the adversary trained on* $n$ *i.i.d. pairs* $(x_i, y_i)$ *by minimizing empirical conditional cross-entropy over a fixed neural-network class. Assume that (A1) $\mathcal{X}$ is finite, and there exists $\gamma > 0$ such that $p(x \mid y) \ge \gamma$ and $q_\theta(x \mid y) \ge \gamma$ for all $x \in \mathcal{X}$ and all $y$ (implemented in practice by softmax clipping) and (A2) the metric satisfies $d(x_i, x_j) \ge d_{\min} > 0$ whenever $x_i \ne x_j$. Then for any fixed pair of candidates $x_i, x_j$ there exists a constant $C > 0$ (depending only on $\gamma$, $d_{\min}$, and the candidate set) such that*

$$\mathbb{E}_Y\left[\left|\mathrm{mPL}(x_i, x_j; Y) - \widetilde{\mathrm{mPL}}(x_i, x_j; Y)\right|\right] \le C\, n^{-\alpha/2} \quad (21)$$

*for all sufficiently large* $n$, *where* $\mathrm{mPL}$ *and* $\widetilde{\mathrm{mPL}}$ *denote the mPL computed using* $p$ *and* $q_\theta$, *respectively. The detailed proof can be found in* **Appendix D.5**.

Proposition 3.3 shows that, as the adversary is trained on more data, its learned posterior $q_\theta(x \mid y)$ converges to the true posterior $p(x \mid y)$ in expected KL, and the mPL computed from $q_\theta$ converges to the true mPL at a polynomial rate. Thus our empirical mPL audit is asymptotically faithful to the true posterior-leakage risk.

*Scope of the audit.* Notably, AmPL's audit is adversary-dependent: for expressive DNN attackers, the exact posterior (and thus exact mPL) is intractable. Instead of pursuing a universal worst-case bound, we adopt a modular framework that supports different threat models by swapping adversary classes. mPL/PBmPL are defined for arbitrary adversaries, and for any fixed class, our sampling audit provides a standard concentration guarantee, i.e., the empirical violation rate converges to the true PBmPL violation probability as the number of samples grows. In practice, we

use high-capacity neural posterior estimators as strong (but approximate) attackers; alternative or stronger attackers can be plugged in and may reveal additional violations. Finally, we focus on a single joint release to a fixed attacker and do not provide a general composition theorem for repeated releases; we position AmPL as an adaptive auditing tool for empirically controlling joint leakage when classical per-user composition is not directly applicable.

**Step ③: Feedback-Driven Perturbation Adjustment.** To balance privacy protection and utility preservation, we adopt an adaptive optimization strategy that iteratively updates the scaling factors $\alpha_1$ and $\alpha_2$ based on adversarial feedback. This adaptation is guided by minimizing a composite loss function $\mathcal{L}(\boldsymbol{\alpha})$, which jointly captures privacy leakage and utility degradation:

$$\mathcal{L}(\boldsymbol{\alpha}) = \lambda_1 \cdot \mathcal{L}_{\text{privacy}}(\boldsymbol{\alpha}) + \lambda_2 \cdot \mathcal{L}_{\text{utility}}(\boldsymbol{\alpha}), \quad (22)$$

where $\lambda_1, \lambda_2 > 0$ are trade-off coefficients that balance the two objectives.

The privacy loss term $\mathcal{L}_{\text{privacy}}(\boldsymbol{\alpha})$ is given by the empirical violation rate $\hat{p}_{\mathcal{S}}$, which estimates the probability that posterior leakage exceeds the privacy budget $\epsilon$ over a sampled set of input-output pairs. The utility loss term $\mathcal{L}_{\text{utility}}(\boldsymbol{\alpha})$ represents the expected semantic distortion caused by the perturbation:

$$\mathcal{L}_{\text{utility}}(\boldsymbol{\alpha}) = \sum_{\ell=1}^{L} \sum_{x \in \mathcal{X}_1} \sum_{y \in \mathcal{Y}} \pi_x \, c_{x,y} \, \Pr\left(\mathcal{M}_\ell(x; \alpha_\ell) = y\right),$$

where $\pi_x$ denotes the prior probability of $x$, and $c_{x,y}$ quantifies the utility loss incurred by reporting $y$ when the true input is $x$.

**Step ④: Bayesian Remapping**. While perturbation mechanisms protect privacy by injecting noise into sensitive data, the resulting outputs may not be optimal for downstream tasks due to semantic distortion. To mitigate this utility loss, we employ *Bayesian remapping*, a post-processing step that refines perturbed outputs based on their inferred posterior distributions. Given a perturbed record $y$, Bayesian remapping selects an alternative output that minimizes the expected utility loss under the posterior, formally defined as:

$$f(y) = \arg\min_{y' \in \mathcal{Y}} \sum_{\ell=1}^{L} \sum_{x \in \mathcal{X}_\ell} \underbrace{\Pr\left[X = x \mid \mathcal{M}_\ell(x; \alpha_\ell) = y\right]}_{\text{posterior of } x \text{ given perturbed record } y} c_{x,y'}. \tag{23}$$

Notably, (Chatzikokolakis et al., 2017) has proved that this transformation preserves the original mDP guarantees under individual perturbed observations. We extend this result by formally proving in Proposition 2.4 that the joint mPL constraint is also preserved under post-processing.

## 4. Case Study: PII/PoII Embedding Protection

We evaluate AmPL on text embeddings because embeddings are widely used in deployed systems and text exhibits lay-

ered sensitivities (PII vs. PoII) that naturally align with level-wise perturbation; it also provides standard threat models (reconstruction/attribute inference) and utility benchmarks (classification/retrieval).[1] Although our experiments perturb in embedding space, mPL/AmPL are modality- and representation-agnostic, requiring only a metric over secret records, a utility loss, and a learned attacker (see Appendix F.8 and Appendix C.3); additional case-study details are deferred to Appendix E.

We use the pre-trained GloVe embeddings (Pennington et al., 2014) and evaluate on three benchmark text datasets: (1) *AG-News* (Zhang et al., 2015), a four-class news classification corpus with 120,000 training and 7,600 test samples (World, Sports, Business, and Sci/Tech); (2) *IMDB* (Maas et al., 2011), a binary sentiment dataset of 50,000 reviews evenly split between positive and negative labels; and (3) the *Amazon Reviews* corpus (Zhang et al., 2015), a large-scale collection spanning multiple domains, containing 34,686,770 reviews from 6,643,669 users over 2,441,053 products.

We use the Exponential Mechanism (EM) for perturbation, which samples directly from a finite candidate set with a distance-aligned utility score, typically yielding high utility for discrete outputs (Feyisetan et al., 2020). In particular, we apply two *adjusted EM* perturbation mechanisms, $\mathcal{M}_{\text{EM}}(\cdot; \alpha_1 \epsilon)$ and $\mathcal{M}_{\text{EM}}(\cdot; \alpha_2 \epsilon)$, to protect PII and PoII, controlled by scaling factors $\alpha_1$ and $\alpha_2$ (with $\alpha_1, \alpha_2 \in [0, 1]$ and $\alpha_1 < \alpha_2$). Formally, for $\ell \in \{1, 2\}$,

$$\Pr\left[\mathcal{M}_{\text{EM}}(x; \alpha_\ell \epsilon) = y\right] = \frac{\exp\left(-\frac{1}{2}\alpha_\ell \epsilon \cdot d_{x,y}\right)}{\sum_{y' \in \mathcal{Y}} \exp\left(-\frac{1}{2}\alpha_\ell \epsilon \cdot d_{x,y'}\right)}, \tag{24}$$

where $\mathcal{M}_1(\cdot)$ (with smaller $\alpha_1$) introduces stronger noise for PII, and $\mathcal{M}_2(\cdot)$ (with larger $\alpha_2$) applies milder noise for PoII.

**Compared methods.** As a baseline, we use the standard EM in Eq. (24) with no level differentiation, i.e., $\alpha_1 = \alpha_2 = 1$. For ablations, we compare our full method (AmPL) against three variants: (i) *AmPL-U* (utility-preserving), which removes identity-salience weighting by setting $\lambda_1 = 0$ in the objective of Eq. (22), thereby optimizing only utility loss; and (ii) *AmPL-P* (privacy-preserving), which sets $\lambda_2 = 0$ in the objective of Eq. (22), thereby optimizing only privacy loss; and (iii) *AmPL-1*, which applies identity-salience weighting to PII only (no PoII weighting).

**Main Results.** Table 1 compares the mPL violation ratio of different perturbation approaches across the three datasets (AG News, IMDB, Amazon) and inference models (RNN, LSTM, Transformer). From the table, we can observe that even with per-record mDP noise (EM), mPL violations remain nontrivial under joint, learned attackers. For example, at $\epsilon = 2.40$ the Transformer attacker flags a sizeable fraction

---

[1]An anonymized artifact is included in the supplementary material.

*Table 1.* Posterior leakage violation ratio (%).

| | RNN | | | LSTM | | | Transformer | | |
|---|---|---|---|---|---|---|---|---|---|
| $\epsilon$ | 2.40 | 2.50 | 2.60 | 2.40 | 2.50 | 2.60 | 2.40 | 2.50 | 2.60 |
| | | | | AG News Dataset | | | | | |
| EM (mDP) | 13.56±0.82 | 14.74±1.30 | 17.79±1.55 | 15.29±2.44 | 16.84±0.72 | 19.47±2.95 | 19.29±1.66 | 20.50±1.38 | 21.75±1.26 |
| AmPL | 8.80±1.30 | 8.34±2.02 | 7.64±0.53 | 10.70±2.66 | 9.83±2.29 | 9.19±0.95 | 15.07±1.10 | 14.91±2.62 | 13.61±2.33 |
| AmPL-U | 10.57±0.86 | 9.81±0.96 | 9.48±1.36 | 11.61±2.15 | 10.51±2.03 | 9.94±1.76 | 16.66±1.63 | 15.68±1.39 | 14.58±1.38 |
| AmPL-P | 8.80±1.32 | 7.94±1.64 | 7.43±0.66 | 10.68±2.61 | 9.71±2.01 | 8.93±1.62 | 15.18±1.63 | 14.91±2.63 | 13.35±2.58 |
| AmPL-1 | 12.46±1.43 | 11.49±1.93 | 9.94±2.18 | 10.79±1.17 | 9.81±2.95 | 9.03±2.65 | 22.10±2.29 | 20.40±2.21 | 18.45±2.79 |
| | | | | IMDB Review Dataset | | | | | |
| EM (mDP) | 11.73±1.01 | 12.42±1.13 | 13.11±2.26 | 8.65±2.06 | 9.14±2.47 | 9.33±2.05 | 12.09±1.36 | 10.87±1.18 | 10.01±0.81 |
| AmPL | 8.71±2.90 | 8.18±1.46 | 7.08±1.14 | 6.90±3.01 | 6.18±3.47 | 7.10±2.89 | 11.42±0.98 | 9.80±0.91 | 9.57±0.36 |
| AmPL-U | 9.92±1.11 | 9.22±1.12 | 8.18±0.84 | 8.14±2.33 | 7.67±2.69 | 6.98±2.24 | 12.07±1.39 | 10.97±1.59 | 9.80±0.81 |
| AmPL-P | 8.67±3.32 | 7.98±0.88 | 6.60±1.13 | 5.83±5.92 | 5.82±4.20 | 6.82±2.16 | 11.65±1.17 | 9.81±0.66 | 9.65±0.68 |
| AmPL-1 | 7.72±0.80 | 7.41±0.45 | 5.75±1.69 | 7.01±0.89 | 6.42±0.83 | 5.79±0.67 | 2.25±1.45 | 1.73±1.29 | 1.06±1.16 |
| | | | | Amazon Review Dataset | | | | | |
| EM (mDP) | 9.30±1.64 | 10.55±0.65 | 12.07±1.75 | 8.52±2.20 | 9.56±0.83 | 10.57±1.62 | 10.78±2.06 | 10.80±2.36 | 12.91±2.18 |
| AmPL | 7.11±2.21 | 6.56±1.13 | 6.28±2.75 | 7.48±2.85 | 6.35±1.77 | 6.22±2.07 | 9.58±4.62 | 8.52±2.90 | 9.09±2.19 |
| AmPL-U | 7.91±0.90 | 7.63±0.95 | 7.34±1.33 | 7.90±0.95 | 7.04±1.16 | 6.60±1.16 | 10.37±1.63 | 9.61±0.87 | 9.72±2.02 |
| AmPL-P | 7.00±1.55 | 6.42±0.94 | 6.50±1.73 | 7.14±2.27 | 6.26±1.32 | 6.15±1.85 | 8.98±3.16 | 8.08±2.21 | 8.68±1.96 |
| AmPL-1 | 7.87±3.82 | 6.86±2.94 | 6.02±2.76 | 7.05±2.87 | 5.57±2.18 | 4.76±1.95 | 8.13±2.53 | 6.69±3.04 | 5.93±3.00 |

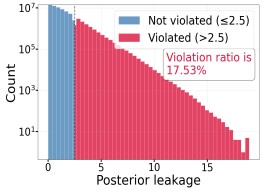

*Figure 4.* Example of mPL distributions derived by a DNN-based inference model (Transformer).

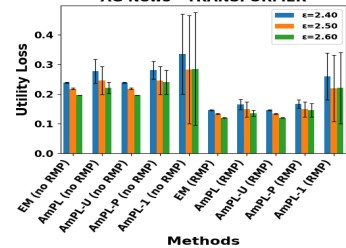

*Figure 5.* Utility loss (using Transformer as inference model).

of violations (e.g., $\approx 12.1\%$ on IMDB), and classical RNNs still expose leakage on AG NEWS (e.g., $\approx 13.6\%$). Figure 4 illustrates the distribution of mPL in the AG News dataset when $\epsilon = 2.50$ and the attacker model is Transformer (more comprehensive results are reported in **Appendix F.1**).

We also observe that EM often shows increasing violation rates as $\epsilon$ grows, which suggests the reduced perturbation at larger $\epsilon$ can dominate and expose more attacker-aligned leakage. Across models, stronger attackers generally lead to higher leakage, especially on AG News where Transformers yield the largest EM violations. On IMDB and Amazon, the orderings differ, but Transformers still remain above LSTMs, highlighting that leakage depends not only on model capacity but also on dataset structure and correlations.

Comparing mechanisms, AmPL consistently achieves the lower violation ratios in all 27 model–dataset–$\epsilon$ settings. Overall, AmPL reduces the violation ratio by 4.13% points on average (roughly 29.89% relative reduction on average), with the largest absolute drop on AG News–LSTM at $\epsilon = 2.60$ (from 19.47% to 9.19%) and the largest relative drop on AG News–RNN at $\epsilon = 2.60$ (from 17.79% to 7.64%, achieves 57.1% relative reduction). Among ablations, AmPL-P is similarly robust, and it outperforms AmPL in most of the settings. AmPL-U, which does not explicitly optimize leakage, yields more moderate gains and performs

similarly to EM. Finally, AmPL-1 (without PoII protection) exhibits highly non-uniform behavior: it can increase violations in some settings, yet dramatically reduces leakage on IMDB–Transformer (down to 1.06 at $\epsilon = 2.60$), indicating strong dataset–attacker interactions when disabling components of AmPL. In all three datasets, using large mPL sampling sizes and setting the achievable PBmPL target to $\tilde{\delta} = 1.05\,\delta^\star$, we obtain astronomically small failure probabilities across all $\epsilon$ (down to $< 10^{-1,000,000}$), indicating violations are effectively impossible at scale; detailed numbers are reported in **Appendix F.2**.

Fig. 5 compares utility loss on AG News under a Transformer adversary (the results of other datasets/inference models are in **Appendix F.3**). Remapping (RMP) substantially improves utility, reducing loss from $\approx 0.22$ to $\approx 0.13$ at $\epsilon = 2.50$ in EM. After remapping, EM, AmPL, AmPL-U and AmPL-P have nearly identical utility (AmPL essentially matches EM), suggesting that RMP dominates by projecting perturbed embeddings back to utility-preserving regions; in contrast, AmPL-1 (RMP) remains notably worse. We also report learning curves for the learned adversary (**Appendix F.6**), showing that attack accuracy saturates quickly with a moderate number of training pairs, while the estimated mPL violation ratio increases as well.

## 5. Conclusions

We formalized metric-normalized posterior leakage and its relaxation (PBmPL), and proposed AmPL, an adaptive, attacker-aligned perturbation mechanism. Across multiple datasets with RNN/LSTM/Transformer, AmPL cuts posterior leakage while keeping utility loss low, yielding a favorable privacy–utility trade-off. Ablations show AmPL-U preserves performance with limited leakage reduction, whereas AmPL-1 boosts protection at modest extra cost. In the future work, we will extend beyond text, strengthen PBmPL composition, and scale to broader adversaries.

## Impact Statements

This work introduces metric-normalized posterior leakage (mPL) and a PBmPL/AmPL auditing-and-repair framework to evaluate and reduce inferential privacy leakage under *joint consumption* of correlated releases. If adopted, these tools could help practitioners uncover privacy failure modes that per-release guarantees may miss, improving privacy evaluation for representation-based ML systems (e.g., embeddings) used in applications such as search, chat, and recommendation. Potential risks include threat-model dependence and misuse. Auditing with learned attackers can underestimate leakage if the adversary class is too weak or misspecified, while malicious actors could repurpose the same methodology to strengthen inference attacks against deployed perturbation schemes. The adaptive loop may also increase computational and environmental costs, and utility-oriented post-processing could affect subpopulations unevenly. We recommend reporting results across multiple attacker families/capacities, treating audit outcomes as conditional rather than universal guarantees, constraining compute budgets, and evaluating robustness across relevant subgroups.

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

# Part I

# Appendix

## Appendix Overview

**Appendix overview.** Appendix A discloses our use of large language models for wording and clarity. Appendix B consolidates key mathematical notation. Appendix C provides additional discussions. Appendix D contains omitted proofs. Appendix E provides additional case-study details. Appendix F reports extended experiments.

## A. The Use of Large Language Models (LLMs)

We used a large language model (LLM) assistant to aid wording, improve clarity, and polish exposition across *Sections 1–5* and *Appendices B–F*. The LLM was not used to generate novel results, code, or citations, and no outputs were accepted without human review. The authors verified the accuracy of all assisted text and take full responsibility for the final content.

## B. Math Notations

*Table 2.* Notation summary. Symbols are grouped by role (spaces/variables, mechanisms/metrics, leakage/certificates, and optimization).

| Symbol | Description |
|---|---|
| **Spaces and random variables** | |
| $\mathcal{X}, \mathcal{Y}$ | Secret (embedding) space and perturbed/release space. |
| $x_i \in \mathcal{X}, y_k \in \mathcal{Y}$ | A candidate secret embedding and a candidate perturbed embedding. |
| $X, X_\ell$ | Random variables for a secret embedding and the $\ell$-th secret in a sequence. |
| $d_{x_i,x_j}$ | Metric distance on $\mathcal{X}$ between $x_i$ and $x_j$. |
| $c_{x_i,y_k}$ | Utility loss when releasing $y_k$ for true input $x_i$. |
| **Mechanisms and parameters** | |
| $\mathcal{M}$ | Perturbation mechanism mapping $\mathcal{X} \to \mathcal{Y}$. |
| $\mathcal{M}(\cdot, \alpha_\ell)\ (\ell = 1, ..., N)$ | Level-wise mechanisms for $N$ sensitivity tiers (e.g., PII vs. PoII). |
| $\boldsymbol{\alpha} = (\alpha_1, \ldots, \alpha_N)$ | Vector of per-level perturbation strengths. |
| $f(y)$ | Bayesian remapping (post-processing) applied to releases. |
| **Leakage and certificates** | |
| $\mathrm{mPL}_{\mathcal{M}}(x_i, x_j; \mathbf{y})$ | Metric-normalized posterior leakage (single/joint; prior→posterior odds change, normalized by $d_{x_i,x_j}$). |
| $\epsilon$ | Target mPL budget (smaller is more private). |
| $p_{\mathcal{X}^2}$ | Violation probability $\Pr[\mathrm{mPL} > \epsilon]$ over random pairs $(x_i, x_j)$ and releases. |
| $\hat{p}_S$ | Empirical estimate of $p_{\mathcal{X}^2}$ from a sampled set $S$ of triples $(x_i, x_j, y)$. |
| $\delta$ | Tolerance on violation frequency for PBmPL. |
| **Optimization and adaptation (AmPL)** | |
| $L(\boldsymbol{\alpha})$ | Composite objective balancing privacy and utility. |
| $L_{\mathrm{privacy}}(\boldsymbol{\alpha})$ | Privacy term (e.g., empirical violation rate). |
| $L_{\mathrm{utility}}(\boldsymbol{\alpha})$ | Expected utility distortion under $\mathcal{M}$. |
| $\lambda_1, \lambda_2$ | Weights trading off privacy vs. utility in $L(\cdot)$. |
| $\eta(t)$ | Learning rate at iteration $t$ for adaptive updates. |
| $\|\boldsymbol{\alpha}^{(t+1)} - \boldsymbol{\alpha}^{(t)}\|_2$ | Step size between consecutive parameter updates. |

## C. Discussions

### C.1. Inference Models based on Explicit Joint Probability

We construct a scenario, in which (1) two secret records $X_1$ and $X_2$ are not independently distributed, (2) observing each perturbed record ($\mathcal{M}_{\mathrm{EM}}(X_1)$ or $\mathcal{M}_{\mathrm{EM}}(X_2)$) *individually* doesn't violate the posterior leakage bound, yet (3) observing $\mathcal{M}_{\mathrm{EM}}(X_1)$ and $\mathcal{M}_{\mathrm{EM}}(X_2)$ *jointly* causes a posterior leakage bound violation for $X_1$.

Suppose that $X_1$ and $X_2$ each take values in $\mathcal{X} = \{x_1, x_2\}$ with the following joint distribution:

$$\Pr(X_1 = x_1, X_2 = x_1) = 0.01, \tag{25}$$
$$\Pr(X_1 = x_1, X_2 = x_2) = 0.49, \tag{26}$$
$$\Pr(X_1 = x_2, X_2 = x_1) = 0.49, \tag{27}$$
$$\Pr(X_1 = x_2, X_2 = x_2) = 0.01. \tag{28}$$

Then each $X_i$ has marginal distribution: $\Pr(X_i = x_1) = 0.5$ and $\Pr(X_i = x_2) = 0.5$. Therefore,

$$\Pr(X_1 = x_i, X_2 = x_j) \neq \Pr(X_1 = x_i)\Pr(X_2 = x_j), \tag{29}$$

$\forall x_i, x_j \in \mathcal{X}$ indicating $X_1$ and $X_2$ are not independent.

We let $\mathcal{M}_{\mathrm{EM}}(X_i) \in \{y_1, y_2\}$. The perturbation probabilities are given by

$$\Pr\left(\mathcal{M}_{\mathrm{EM}}(X_i) = y_1 \mid X_i = x_1\right) = 0.72, \tag{30}$$
$$\Pr\left(\mathcal{M}_{\mathrm{EM}}(X_i) = y_2 \mid X_i = x_1\right) = 0.28, \tag{31}$$
$$\Pr\left(\mathcal{M}_{\mathrm{EM}}(X_i) = y_1 \mid X_i = x_2\right) = 0.28, \tag{32}$$
$$\Pr\left(\mathcal{M}_{\mathrm{EM}}(X_i) = y_2 \mid X_i = x_2\right) = 0.72. \tag{33}$$

Finally, we set the privacy budget $\epsilon = 1$.

**(1) The posterior by observing each individual perturbed record**: The *posterior odds* given the observation $\mathcal{M}_{\mathrm{EM}}(X_i) = y_1$ is

$$\left| \ln\left( \frac{\Pr(X_i = x_1 \mid \mathcal{M}_{\mathrm{EM}}(X_i) = y_1)}{\Pr(X_i = x_2 \mid \mathcal{M}_{\mathrm{EM}}(X_i) = y_1)} \Big/ \frac{\Pr(X_i = x_1)}{\Pr(X_i = x_2)} \right) \right| = \left| \ln\left( \frac{\Pr(\mathcal{M}_{\mathrm{EM}}(X_i) = y_1 \mid X_i = x_1)}{\Pr(\mathcal{M}_{\mathrm{EM}}(X_i) = y_1 \mid X_i = x_2)} \right) \right| \tag{34}$$

$$= \left| \ln\left( \frac{0.72}{0.28} \right) \right| \tag{35}$$

$$= 0.9444 \tag{36}$$

$$< \epsilon. \tag{37}$$

Similarly, the *posterior odds* given the observation $\mathcal{M}_{\mathrm{EM}}(X_i) = y_2$ is

$$\left| \ln\left( \frac{\Pr(X_i = x_1 \mid \mathcal{M}_{\mathrm{EM}}(X_i) = y_2)}{\Pr(X_i = x_2 \mid \mathcal{M}_{\mathrm{EM}}(X_i) = y_2)} \Big/ \frac{\Pr(X_i = x_1)}{\Pr(X_i = x_2)} \right) \right| = \left| \ln\left( \frac{\Pr(\mathcal{M}_{\mathrm{EM}}(X_i) = y_2 \mid X_i = x_1)}{\Pr(\mathcal{M}_{\mathrm{EM}}(X_i) = y_2 \mid X_i = x_2)} \right) \right| \tag{38}$$

$$= \left| \ln\left( \frac{0.28}{0.72} \right) \right| \tag{39}$$

$$= 0.9444 \tag{40}$$

$$< \epsilon. \tag{41}$$

indicating that $\left| \ln\left( \frac{\Pr(X_i = x_1 \mid \mathcal{M}_{\mathrm{EM}}(X_i) = y_j)}{\Pr(X_i = x_2 \mid \mathcal{M}_{\mathrm{EM}}(X_i) = y_j)} \Big/ \frac{\Pr(X_i = x_1)}{\Pr(X_i = x_2)} \right) \right| \leq 1$ for each $X_i$ and $y_k$.

**(2) Posterior leakage give joint observation**: The *posterior odds* given the joint observation $\mathcal{M}_{\mathrm{EM}}(X_1) =$

$y_1, \mathcal{M}_{\mathrm{EM}}(X_2) = y_2$ is

$$\left| \ln \left( \frac{\Pr(X_1 = x_1 \mid \mathcal{M}_{\mathrm{EM}}(X_1) = y_1, \mathcal{M}_{\mathrm{EM}}(X_2) = y_2)}{\Pr(X_1 = x_2 \mid \mathcal{M}_{\mathrm{EM}}(X_1) = y_1, \mathcal{M}_{\mathrm{EM}}(X_2) = y_2)} \middle/ \frac{\Pr(X_1 = x_1)}{\Pr(X_1 = x_2)} \right) \right| \tag{42}$$

$$= \left| \ln \left( \frac{\Pr(X_1 = x_1 \mid \mathcal{M}_{\mathrm{EM}}(X_1) = y_1, \mathcal{M}_{\mathrm{EM}}(X_2) = y_2)}{\Pr(X_1 = x_2 \mid \mathcal{M}_{\mathrm{EM}}(X_1) = y_1, \mathcal{M}_{\mathrm{EM}}(X_2) = y_2)} \middle/ 1 \right) \right| \tag{43}$$

$$= \left| \ln \left( \frac{\Pr(X_1 = x_1, \mathcal{M}_{\mathrm{EM}}(X_1) = y_1, \mathcal{M}_{\mathrm{EM}}(X_2) = y_2)}{\Pr(X_1 = x_2, \mathcal{M}_{\mathrm{EM}}(X_1) = y_1, \mathcal{M}_{\mathrm{EM}}(X_2) = y_2)} \right) \right| \tag{44}$$

$$= \left| \ln \left( \frac{\sum_{x \in \{x_1, x_2\}} \Pr(X_1 = x_1, X_2 = x, \mathcal{M}_{\mathrm{EM}}(X_1) = y_1, \mathcal{M}_{\mathrm{EM}}(X_2) = y_2)}{\sum_{x \in \{x_1, x_2\}} \Pr(X_1 = x_2, X_2 = x, \mathcal{M}_{\mathrm{EM}}(X_1) = y_1, \mathcal{M}_{\mathrm{EM}}(X_2) = y_2)} \right) \right| \tag{45}$$

$$= \left| \ln \left( \frac{\sum_{x \in \{x_1, x_2\}} \Pr(X_1 = x_1, X_2 = x) \Pr(\mathcal{M}_{\mathrm{EM}}(X_1) = y_1, \mathcal{M}_{\mathrm{EM}}(X_2) = y_2 | X_1 = x_1, X_2 = x)}{\sum_{x \in \{x_1, x_2\}} \Pr(X_1 = x_2, X_2 = x) \Pr(\mathcal{M}_{\mathrm{EM}}(X_1) = y_1, \mathcal{M}_{\mathrm{EM}}(X_2) = y_2 | X_1 = x_2, X_2 = x)} \right) \right|$$

$$= \left| \ln \left( \frac{\sum_{x \in \{x_1, x_2\}} \Pr(X_1 = x_1, X_2 = x) \Pr(\mathcal{M}_{\mathrm{EM}}(X_1) = y_1 | X_1 = x_1) \Pr(\mathcal{M}_{\mathrm{EM}}(X_2) = y_2 | X_2 = x)}{\sum_{x \in \{x_1, x_2\}} \Pr(X_1 = x_2, X_2 = x) \Pr(\mathcal{M}_{\mathrm{EM}}(X_1) = y_1 | X_1 = x_2) \Pr(\mathcal{M}_{\mathrm{EM}}(X_2) = y_2 | X_2 = x)} \right) \right|$$

$$= \left| \ln \left( \frac{0.01 \times 0.72 \times 0.28 + 0.49 \times 0.72 \times 0.72}{0.49 \times 0.28 \times 0.28 + 0.01 \times 0.28 \times 0.72} \right) \right| \tag{46}$$

$$= 1.8456 \tag{47}$$

$$> \epsilon. \tag{48}$$

$$\left| \ln \left( \frac{\Pr(X_1 = x_1 \mid \mathcal{M}_{\mathrm{EM}}(X_1) = y_1, \mathcal{M}_{\mathrm{EM}}(X_2) = y_2)}{\Pr(X_1 = x_2 \mid \mathcal{M}_{\mathrm{EM}}(X_1) = y_1, \mathcal{M}_{\mathrm{EM}}(X_2) = y_2)} \middle/ \frac{\Pr(X_1 = x_1)}{\Pr(X_1 = x_2)} \right) \right|$$

$$= \left| \ln \left( \frac{\Pr(X_1 = x_1 \mid \mathcal{M}_{\mathrm{EM}}(X_1) = y_1, \mathcal{M}_{\mathrm{EM}}(X_2) = y_2)}{\Pr(X_1 = x_2 \mid \mathcal{M}_{\mathrm{EM}}(X_1) = y_1, \mathcal{M}_{\mathrm{EM}}(X_2) = y_2)} \right) \right|$$

$$= \left| \ln \left( \frac{\Pr(X_1 = x_1, \mathcal{M}_{\mathrm{EM}}(X_1) = y_1, \mathcal{M}_{\mathrm{EM}}(X_2) = y_2)}{\Pr(X_1 = x_2, \mathcal{M}_{\mathrm{EM}}(X_1) = y_1, \mathcal{M}_{\mathrm{EM}}(X_2) = y_2)} \right) \right|$$

$$= \left| \ln \left( \frac{\sum_{x \in \{x_1, x_2\}} \Pr(X_1 = x_1, X_2 = x, \mathcal{M}_{\mathrm{EM}}(X_1) = y_1, \mathcal{M}_{\mathrm{EM}}(X_2) = y_2)}{\sum_{x \in \{x_1, x_2\}} \Pr(X_1 = x_2, X_2 = x, \mathcal{M}_{\mathrm{EM}}(X_1) = y_1, \mathcal{M}_{\mathrm{EM}}(X_2) = y_2)} \right) \right|$$

$$= \left| \ln \left( \frac{\sum_{x \in \{x_1, x_2\}} \Pr(X_1 = x_1, X_2 = x) \Pr(\mathcal{M}_{\mathrm{EM}}(X_1) = y_1, \mathcal{M}_{\mathrm{EM}}(X_2) = y_2 \mid X_1 = x_1, X_2 = x)}{\sum_{x \in \{x_1, x_2\}} \Pr(X_1 = x_2, X_2 = x) \Pr(\mathcal{M}_{\mathrm{EM}}(X_1) = y_1, \mathcal{M}_{\mathrm{EM}}(X_2) = y_2 \mid X_1 = x_2, X_2 = x)} \right) \right|$$

$$= \left| \ln \left( \frac{\sum_{x \in \{x_1, x_2\}} \Pr(X_1 = x_1, X_2 = x) \Pr(\mathcal{M}_{\mathrm{EM}}(X_1) = y_1 \mid X_1 = x_1) \Pr(\mathcal{M}_{\mathrm{EM}}(X_2) = y_2 \mid X_2 = x)}{\sum_{x \in \{x_1, x_2\}} \Pr(X_1 = x_2, X_2 = x) \Pr(\mathcal{M}_{\mathrm{EM}}(X_1) = y_1 \mid X_1 = x_2) \Pr(\mathcal{M}_{\mathrm{EM}}(X_2) = y_2 \mid X_2 = x)} \right) \right|$$

$$= \left| \ln \left( \frac{0.01 \cdot 0.72 \cdot 0.28 + 0.49 \cdot 0.72 \cdot 0.72}{0.49 \cdot 0.28 \cdot 0.28 + 0.01 \cdot 0.28 \cdot 0.72} \right) \right|$$

$$= \left| \ln \left( \frac{0.256032}{0.040432} \right) \right|$$

$$= 1.8456 > \epsilon \tag{49}$$

## C.2. Per-User Accounting As a Complementary Mitigation

We note that when records can be cleanly grouped by user and the mechanism is explicitly designed with per-user accounting, a per-user privacy budget is a natural and effective way to mitigate composition across correlated records. In the classical DP setting, this corresponds to treating each user as the "unit of protection," ensuring that all contributions from the same user share a fixed budget.

However, *user grouping is not always known or reliable in the kinds of applications we target.* For example, in many text embedding-based systems, the mechanism does not have a trusted user identifier for each record. Posts may come from multiple accounts controlled by the same person, or a single account may refer to several different individuals or secrets. Determining that a set of words, embeddings, or snippets describe the same person or the same underlying secret is itself part of the adversarial inference task, for example, linking posts across accounts, or linking mentions of the same individual across different documents. In such settings, a per-user budget implicitly assumes that this partition into users is known and

enforced by the defender, whereas our threat model explicitly allows the adversary to aggregate any correlated releases they can link. This issue is also reflected in our case study. The public datasets we use do not contain user identifiers or reliable group labels that could serve as a ground truth for "per-user" segmentation. Constructing a per-user baseline would therefore require introducing additional, task-specific grouping heuristics (e.g., clustering by text similarity), which are orthogonal to our core threat model. Since our goal is to study joint leakage over arbitrary correlated secrets, without assuming that the defender knows how records should be grouped, we deliberately adopt a user-agnostic formulation of mPL and AmPL.

On the other hand, we see per-user budgeting as a complementary mitigation, not as an alternative to our framework. In applications where reliable user identifiers and grouping assumptions are available, our mechanisms and audits can be combined with per-user accounting: the system can enforce a per-user privacy budget while our framework still evaluates whether correlated secrets within or across those groups violate the intended metric privacy guarantees.

### C.3. Perturbation Space: Embeddings vs. Text.

Our current implementation instantiates AmPL by perturbing word embeddings, because embeddings are a common interface in modern NLP systems (e.g., for search, retrieval, and recommendation). However, the *framework* itself is agnostic to whether perturbations are applied in embedding space or directly on text.

Formally, let $\mathcal{X}$ denote the space of original text (words or sentences) and $\mathcal{Y}$ the space of perturbed outputs (which may be text or embeddings). Any defense mechanism that specifies a randomized map $\mathcal{M} : \mathcal{X} \to \mathrm{Distr}(\mathcal{Y}), \quad x \mapsto \mathcal{M}(\cdot \mid x)$ induces a perturbation method and a corresponding posterior $\Pr[X \mid Y]$. Our mPL definition and learned-adversary attack are defined purely in terms of this posterior, and therefore apply unchanged to *any* such stochastic channel.

In particular, defenses that act directly on text, such as token deletion, insertion of noise characters, or synonym substitution, still define a stochastic mapping from original text $x$ to perturbed text $y$. An attacker observing $y$ can then process it through the same embedding model (or any other feature extractor) and train a predictor exactly as in our experiments. From the perspective of mPL and the learned adversary, the only requirement is that $(X, Y)$ are jointly distributed via some randomized mechanism; the choice of operating in embedding space or text space is an implementation detail of the defense, not a limitation of the framework.

# D. Omitted Proofs

### D.1. Proof of Proposition 2.4 (Post-processing for Bounded mPL)

**Proposition 1. (Post-processing for bounded mPL)** Let $\mathcal{M} : \mathcal{X} \to \mathcal{Y}$ be a randomized mechanism that satisfies the bounded joint mPL constraint. For any (possibly randomized) function $f : \mathcal{Y} \to \mathcal{Z}$, define the post-processed mechanism $(f \circ \mathcal{M})(x) \triangleq f(\mathcal{M}(x))$, $\forall x \in \mathcal{X}$, where $\mathcal{Z} = \mathrm{Range}(f \circ \mathcal{M})$. Then $f \circ \mathcal{M}$ also satisfies the bounded joint mPL constraint:

$$\sup_{x_i \neq x_j} \sup_{\mathbf{z} \in \mathcal{Z}^L} \mathrm{mPL}_{f \circ \mathcal{M}}(x_i, x_j, \mathbf{z}) \leq \epsilon. \tag{50}$$

*Proof.* Let $f^{-1}(\mathbf{z}) = \{\mathbf{y} : f(\mathbf{y}) = \mathbf{z}\}$ denote the preimage of $z$. Fix any pair $x_i, x_j \in \mathcal{X}$ and any $\mathbf{y} \in \mathcal{Y}^L$, we have

$$\mathrm{mPL}_{f \circ \mathcal{M}}(x_i, x_j, \mathbf{y}) \tag{51}$$

$$= \ln\left(\frac{\Pr(X_\ell = x_i \mid \{f \circ \mathcal{M}(X_1), \dots, f \circ \mathcal{M}(X_L)\} = \mathbf{z})}{\Pr(X_\ell = x_j \mid \{f \circ \mathcal{M}(X_1), \dots, f \circ \mathcal{M}(X_L)\} = \mathbf{z})} \middle/ \frac{\Pr(X_\ell = x_i)}{\Pr(X_\ell = x_j)}\right) \tag{52}$$

$$= \ln\left(\frac{\Pr(X_\ell = x_i, \{\mathcal{M}(X_1), \dots, \mathcal{M}(X_L)\} = f^{-1}(\mathbf{z}))}{\Pr(X_\ell = x_j, \{\mathcal{M}(X_1), \dots, \mathcal{M}(X_L)\} = f^{-1}(\mathbf{z}))} \middle/ \frac{\Pr(X_\ell = x_i)}{\Pr(X_\ell = x_j)}\right) \tag{53}$$

$$\leq d_{x_i, x_j} \epsilon. \tag{54}$$

$\square$

### D.2. Proof of Proposition 2.5 (Single-Observation Equivalence of mPL and mDP)

**Proposition 2** (Single-observation equivalence of mPL and mDP)**.** Let $\mathcal{M}$ be a perturbation mechanism on a metric secret space $(\mathcal{X}, d)$. Define the single-observation mPL for a pair $(x_i, x_j)$ and observation $y$ by

$$\mathrm{mPL}_{\mathcal{M}}((x_i, x_j), y) = \frac{1}{d_{x_i, x_j}} \left| \ln \frac{\Pr(X = x_i \mid \mathcal{M}(X) = y)}{\Pr(X = x_j \mid \mathcal{M}(X) = y)} - \ln \frac{\Pr(X = x_i)}{\Pr(X = x_j)} \right|. \tag{55}$$

For any $\epsilon \geq 0$, $\mathcal{M}$ satisfies $(\epsilon, d)$-mDP *if and only if* the single-observation mPL bound holds, i.e., $\sup_{x_i \neq x_j} \sup_{y \in \mathcal{Y}} \mathrm{mPL}_{\mathcal{M}}((x_i, x_j), y) \leq \epsilon$.

*Proof.* Fix any pair $\Pr(X = x_i), \Pr(X = x_j) > 0$. By Bayes' rule,

$$\frac{\Pr[X = x_i \mid \mathcal{M}(X) = y]}{\Pr[X = x_j \mid \mathcal{M}(X) = y]} = \frac{\Pr[\mathcal{M}(X) = y \mid X = x_i]}{\Pr[\mathcal{M}(X) = y \mid X = x_j]} \cdot \frac{\Pr(X = x_i)}{\Pr(X = x_j)}. \tag{56}$$

$$\Leftrightarrow \quad \ln \frac{\Pr[X = x_i \mid \mathcal{M}(X) = y]}{\Pr[X = x_j \mid \mathcal{M}(X) = y]} - \ln \frac{\Pr(X = x_i)}{\Pr(X = x_j)} = \ln \frac{\Pr[\mathcal{M}(X) = y \mid X = x_i]}{\Pr[\mathcal{M}(X) = y \mid X = x_j]}. \tag{57}$$

Therefore

$$\underbrace{\frac{1}{d_{x_i, x_j}} \left| \ln \frac{\Pr(X = x_i \mid \mathcal{M}(X) = y)}{\Pr(X = x_j \mid \mathcal{M}(X) = y)} - \ln \frac{\Pr(X = x_i)}{\Pr(X = x_j)} \right| \leq \epsilon}_{\text{Pointwise form of bounded mPL constraint}} \tag{58}$$

$$\Leftrightarrow \quad \underbrace{\ln \frac{\Pr[\mathcal{M}(X) = y \mid X = x_i]}{\Pr[\mathcal{M}(X) = y \mid X = x_j]} \leq \epsilon \, d_{x_i, x_j}}_{\text{Pointwise form of mDP}}. \tag{59}$$

which concludes the proof. $\qquad\square$

### D.3. Proof of Proposition 2.6 (Independent-Observation Equivalence of mPL and mDP)

**Proposition 3** (Independent-observation equivalence of mPL and mDP). If the $L$ secret words $X_1, \ldots, X_L$ are independently distributed, then ensuring $\sup_{x_i \neq x_j} \sup_{y^\ell \in \mathcal{Y}} \mathrm{mPL}_{\mathcal{M}}((x_i, x_j), y^\ell) \leq \epsilon$ for each $y^\ell$ ($\ell = 1, ..., L$) is sufficient to guarantee $\sup_{x_i \neq x_j} \sup_{\mathbf{y} \in \mathcal{Y}^L} \mathrm{mPL}_{\mathcal{M}}(x_i, x_j, \mathbf{y}) \leq \epsilon$.

*Proof.* First, if the random variables $(X_\ell, Z_l)$ are independent with $(X_t, Z_t)$, and $\tilde{\mathcal{M}}$ is a mearable function, then $\tilde{\mathcal{M}}(X_\ell, Z_l)$ and $\tilde{\mathcal{M}}(X_t, Z_t)$ are independent, as measurable functions preserve independence in probability theory [reference].

Then, we can obtain

$$\Pr \left[ X_\ell = x \mid \left\{ \tilde{\mathcal{M}}(X_1, Z), \ldots, \tilde{\mathcal{M}}(X_L, Z) \right\} = \mathbf{y} \right] \tag{60}$$

$$= \frac{\Pr \left[ X_\ell = x, \left\{ \tilde{\mathcal{M}}(X_1, Z), \ldots, \tilde{\mathcal{M}}(X_L, Z) \right\} = \mathbf{y} \right]}{\Pr \left[ \left\{ \tilde{\mathcal{M}}(X_1, Z), \ldots, \tilde{\mathcal{M}}(X_L, Z) \right\} = \mathbf{y} \right]} \tag{61}$$

$$= \frac{\prod_{t=1, t \neq \ell}^{L} \Pr \left[ \tilde{\mathcal{M}}(X_t, Z) = y_t \right] \Pr \left[ X_\ell = x, \tilde{\mathcal{M}}(X_\ell, Z) = y_\ell \right]}{\prod_{t=1}^{L} \Pr \left[ \tilde{\mathcal{M}}(X_t, Z) = y_t \right]} \tag{62}$$

$$= \Pr \left[ X_\ell = x \, \middle| \, \tilde{\mathcal{M}}(X_\ell, Z) = y_\ell \right] \tag{63}$$

$$\square$$

### D.4. Proof of Proposition 3.2 (Concentration Guarantees)

**Proposition 4.** [Concentration Guarantee for Probabilistic mDP Sampling] If the empirical violation rate satisfies $\hat{p}_{\mathcal{S}_\ell} = \xi \delta$ for some constant $\xi < 1$, then $\Pr \left[ p_{\mathcal{X}_\ell^2} \leq \delta \right] \geq 1 - 2 \exp(-2 S_\ell (1 - \xi)^2 \delta^2)$.

*Proof.* Suppose the empirical violation rate satisfies $\hat{p}_{\mathcal{S}} = \xi \delta$ for some constant $\xi < 1$. Our goal is to show that, with high probability, the true violation probability $p_{\mathcal{X}^2}$ is at most $\delta$.

First, by Hoeffding's inequality, for any $t > 0$:

$$\Pr \left[ |\hat{p}_{\mathcal{S}} - p_{\mathcal{X}^2}| > t \right] \leq 2 e^{-2 S_\ell t^2}. \tag{64}$$

Let $t = (1 - \xi)\delta$. Then:

$$\Pr\left[|\hat{p}_{\mathcal{S}} - p_{\mathcal{X}^2}| > (1 - \xi)\delta\right] \leq 2e^{-2S_\ell(1-\xi)^2\delta^2}. \tag{65}$$

This implies:

$$\Pr\left[|\hat{p}_{\mathcal{S}} - p_{\mathcal{X}^2}| \leq (1 - \xi)\delta\right] \geq 1 - 2e^{-2S_\ell(1-\xi)^2\delta^2}. \tag{66}$$

Now, under the event that $|\hat{p}_{\mathcal{S}} - p_{\mathcal{X}^2}| \leq (1 - \xi)\delta$, we have:

$$p_{\mathcal{X}^2} \leq \hat{p}_{\mathcal{S}} + (1 - \xi)\delta = \xi\delta + (1 - \xi)\delta = \delta. \tag{67}$$

Thus:

$$\Pr\left[p_{\mathcal{X}^2} \leq \delta\right] \geq 1 - 2e^{-2S_\ell(1-\xi)^2\delta^2}, \tag{68}$$

which completes the proof. $\qquad\square$

### D.5. Proof of Proposition 3.3 (Asymptotic Faithfulness of the mPL Audit)

**Proposition 5** (Asymptotic faithfulness of the mPL audit). *Let $p(x \mid y)$ denote the true posterior and $q_\theta(x \mid y)$ the adversary trained on $n$ i.i.d. pairs $(x_i, y_i)$ by minimizing empirical conditional cross-entropy over a fixed neural-network class. Assume that (A1) $\mathcal{X}$ is finite, and there exists $\gamma > 0$ such that $p(x \mid y) \geq \gamma$ and $q_\theta(x \mid y) \geq \gamma$ for all $x \in \mathcal{X}$ and all $y$ (implemented in practice by softmax clipping) and (A2) the metric satisfies $d(x_i, x_j) \geq d_{\min} > 0$ whenever $x_i \neq x_j$. Then for any fixed pair of candidates $x_i, x_j$ there exists a constant $C > 0$ (depending only on $\gamma$, $d_{\min}$, and the candidate set) such that*

$$\mathbb{E}_Y\left[\left|\mathrm{mPL}(x_i, x_j; Y) - \widetilde{\mathrm{mPL}}(x_i, x_j; Y)\right|\right] \leq C\, n^{-\alpha/2} \tag{69}$$

*for all sufficiently large $n$, where $\mathrm{mPL}$ and $\widetilde{\mathrm{mPL}}$ denote the mPL computed using $p$ and $q_\theta$, respectively.*

**Lemma D.1** (Cross-entropy and expected KL). *Define the population cross-entropy loss*

$$\mathcal{L}(\theta) := \mathbb{E}_{(X,Y)}\left[-\log q_\theta(X \mid Y)\right]. \tag{70}$$

*Then*

$$\mathcal{L}(\theta) = \mathbb{E}_Y\left[H\big(p(\cdot \mid Y)\big)\right] + \mathbb{E}_Y\left[\mathrm{KL}\big(p(\cdot \mid Y) \,\|\, q_\theta(\cdot \mid Y)\big)\right], \tag{71}$$

*where $H\big(p(\cdot \mid Y)\big)$ is the conditional entropy of the true posterior, therefore minimizing $\mathcal{L}(\theta)$ is equivalent (up to an additive constant) to minimizing the expected posterior KL divergence $\mathbb{E}_Y[\mathrm{KL}(p(\cdot \mid Y) \,\|\, q_\theta(\cdot \mid Y))]$.*

*Proof.* For any fixed $y$,

$$\mathbb{E}_{X|Y=y}\left[-\log q_\theta(X \mid y)\right] = \sum_x p(x \mid y)\big(-\log q_\theta(x \mid y)\big). \tag{72}$$

Add and subtract $\log p(x \mid y)$:

$$\sum_x p(x \mid y)\big(-\log q_\theta(x \mid y)\big) = \sum_x p(x \mid y)\big(-\log p(x \mid y)\big) + \sum_x p(x \mid y) \log \frac{p(x \mid y)}{q_\theta(x \mid y)}$$
$$= H\big(p(\cdot \mid y)\big) + \mathrm{KL}\big(p(\cdot \mid y) \,\|\, q_\theta(\cdot \mid y)\big).$$

Taking expectation over $Y$ yields the claim:

$$\mathcal{L}(\theta) = \mathbb{E}_Y\left[H\big(p(\cdot \mid Y)\big)\right] + \mathbb{E}_Y\left[\mathrm{KL}\big(p(\cdot \mid Y) \,\|\, q_\theta(\cdot \mid Y)\big)\right]. \tag{73}$$
$\square$

**Lemma D.2** (Lipschitz stability of mPL w.r.t. posterior). *Suppose Assumptions A1 and A2 hold. Then for any $x_i, x_j, y$,*

$$\left|\mathrm{mPL}(x_i, x_j; y) - \widetilde{\mathrm{mPL}}(x_i, x_j; y)\right| \leq \frac{2}{\gamma d_{\min}}\big\|p(\cdot \mid y) - q_\theta(\cdot \mid y)\big\|_1. \tag{74}$$

*Proof.* By definition, the posterior-dependent part of mPL is a log-odds term of the form

$$g(a, b) := \log \frac{a}{b},\tag{75}$$

where $a = p(x_i \mid y)$ and $b = p(x_j \mid y)$ for $\mathrm{mPL}$, and $a' = q_\theta(x_i \mid y)$, $b' = q_\theta(x_j \mid y)$ for $\widetilde{\mathrm{mPL}}$.

On the domain $[\gamma, 1 - \gamma]^2$, the gradient is

$$\nabla g(a, b) = \left( \frac{1}{a}, -\frac{1}{b} \right),\tag{76}$$

and by Assumption **A1**

$$\left| \frac{1}{a} \right| \le \frac{1}{\gamma}, \qquad \left| \frac{1}{b} \right| \le \frac{1}{\gamma}.\tag{77}$$

Thus the $\ell_1$-norm of the gradient is bounded:

$$\|\nabla g(a, b)\|_1 = \left| \frac{1}{a} \right| + \left| \frac{1}{b} \right| \le \frac{2}{\gamma}.\tag{78}$$

By the mean value theorem, this implies a Lipschitz bound

$$\left| g(a, b) - g(a', b') \right| \le \frac{2}{\gamma} \left\| (a, b) - (a', b') \right\|_1.\tag{79}$$

Applying this with

$$a = p(x_i \mid y), \quad b = p(x_j \mid y), \quad a' = q_\theta(x_i \mid y), \quad b' = q_\theta(x_j \mid y),\tag{80}$$

and using the lower bound on the metric distance $d(x_i, x_j) \ge d_{\min} > 0$ from Assumption **A2**, we obtain

$$\left| \mathrm{mPL}(x_i, x_j; y) - \widetilde{\mathrm{mPL}}(x_i, x_j; y) \right| \le \frac{2}{\gamma d_{\min}} \left\| (a, b) - (a', b') \right\|_1.\tag{81}$$

Finally, since $(a, b)$ and $(a', b')$ are sub-vectors of $p(\cdot \mid y)$ and $q_\theta(\cdot \mid y)$, we have

$$\left\| (a, b) - (a', b') \right\|_1 \le \left\| p(\cdot \mid y) - q_\theta(\cdot \mid y) \right\|_1,\tag{82}$$

which gives the claimed bound. $\qquad \square$

*Proof of Proposition 3.3.* In practice, we train $q_\theta$ by minimizing the empirical conditional cross-entropy

$$\hat{\mathcal{L}}(\theta) = \frac{1}{n} \sum_{i=1}^{n} -\log q_\theta(x_i \mid y_i),\tag{83}$$

which is a Monte Carlo estimate of the population loss

$$\mathcal{L}(\theta) = \mathbb{E}_{(X, Y)} \left[ -\log q_\theta(X \mid Y) \right].\tag{84}$$

Generalization bounds for deep networks trained with cross-entropy, based on spectral norms and margins, show that with high probability over the draw of the training sample, the excess population cross-entropy of the trained model over the best-in-class predictor decays at a polynomial rate in $n$ (see, e.g., (Bartlett et al., 2017)). Combined with Lemma D.1, this implies that there exist constants $C_0, \alpha > 0$ such that, for sufficiently large $n$,

$$\mathbb{E}_Y \left[ \mathrm{KL}\big(p(\cdot \mid Y) \,\|\, q_\theta(\cdot \mid Y)\big) \right] \le C_0 \, n^{-\alpha}.\tag{85}$$

Next, by Lemma D.2 and Pinsker's inequality,

$$\left\| p(\cdot \mid y) - q_\theta(\cdot \mid y) \right\|_1 \le \sqrt{2 \, \mathrm{KL}\big(p(\cdot \mid y) \,\|\, q_\theta(\cdot \mid y)\big)},\tag{86}$$

we obtain, for each $y$,

$$\left| \mathrm{mPL}(x_i, x_j; y) - \widetilde{\mathrm{mPL}}(x_i, x_j; y) \right| \leq \frac{2\sqrt{2}}{\gamma d_{\min}} \sqrt{\mathrm{KL}\big(p(\cdot \mid y) \,\|\, q_\theta(\cdot \mid y)\big)}. \tag{87}$$

Taking expectation over $Y$ and applying Jensen's inequality to the concave function $z \mapsto \sqrt{z}$ yields

$$\mathbb{E}_Y\left[ \left| \mathrm{mPL}(x_i, x_j; Y) - \widetilde{\mathrm{mPL}}(x_i, x_j; Y) \right| \right] \leq \frac{2\sqrt{2}}{\gamma d_{\min}} \sqrt{\mathbb{E}_Y\left[ \mathrm{KL}\big(p(\cdot \mid Y) \,\|\, q_\theta(\cdot \mid Y)\big) \right]}. \tag{88}$$

Using the generalization bound $\mathbb{E}_Y[\mathrm{KL}(p(\cdot \mid Y) \,\|\, q_\theta(\cdot \mid Y))] \leq C_0 n^{-\alpha}$ then gives

$$\mathbb{E}_Y\left[ \left| \mathrm{mPL}(x_i, x_j; Y) - \widetilde{\mathrm{mPL}}(x_i, x_j; Y) \right| \right] \leq \frac{2\sqrt{2C_0}}{\gamma d_{\min}}\, n^{-\alpha/2}. \tag{89}$$

Setting $C = 2\sqrt{2C_0}/(\gamma d_{\min})$ proves (69) and the proposition. $\qquad\square$

## E. Additional Details of Case Study

In this section, we provide additional details for the case study, including the design of the data perturbation and utiltiy loss, serving as complementary material to Section 4 of the main paper.

### E.1. 2-Level Data Perturbation

Let $\mathcal{U}$ denote the complete set of word embeddings in the dataset. We define $\mathcal{X}_1$ and $\mathcal{X}_2$ as the subsets corresponding to PII and PoII embeddings, respectively. The overall set of sensitive embeddings is given by $\mathcal{X} = \mathcal{X}_1 \cup \mathcal{X}_2$, which is a subset of the full embedding set, i.e., $\mathcal{X} \subseteq \mathcal{U}$. To identify clear instances of PII, we apply *Named Entity Recognition (NER)* using the spaCy Python library[2]. Specifically, we assign a Level 1 privacy label to any word token identified by the NER model as a likely PII entity, such as `PERSON`, `GPE`, or `ORG`. These Level 1 tokens form the subset $\mathcal{X}_1 \subset \mathcal{U}$.

For the complementary set $\mathcal{U} \setminus \mathcal{X}_1$, we adopt the cosine-similarity-based approach proposed by Hassan et al. (Hassan et al., 2023) to identify *Potentially Identifiable Information (PoII)*. Specifically, we compute the cosine similarity between each token in $\mathcal{U} \setminus \mathcal{X}_1$ and the tokens in $\mathcal{X}_1$ using pretrained GloVe embeddings. The top 10% of tokens from $\mathcal{U} \setminus \mathcal{X}_1$ with the highest similarity scores are labeled as Level 2 privacy and form the PoII set $\mathcal{X}_2$. All remaining tokens, those not classified as either PII or PoII, are treated as non-sensitive.

### E.2. Utility Loss Calculation

The *utility loss* quantifies the impact of the obfuscation process on sentence quality. For a sensitive token embedding $x_i$ replaced by a perturbed embedding $y_k$, the loss is defined as

$$c_{i,k} = 1 - \frac{Sim(x_i, y_k) + 1}{2}, \tag{90}$$

where $Sim(x_i, y_k)$ denotes the cosine similarity between $x_i$ and $y_k$: $Sim(x_i, y_k) = \frac{x_i \cdot y_k}{\|x_i\|\|y_k\|}$. The normalization term $\frac{(\cdot)+1}{2}$ maps the cosine similarity from the range $[-1, 1]$ to $[0, 1]$, ensuring that $c_{i,k} \in [0, 1]$. Thus, $c_{i,k}$ captures the semantic deviation introduced by the perturbation, with higher values indicating greater semantic loss. Each token embedding is represented using pre-trained 100-dimensional GloVe vectors, which preserve the structure and context of the original sentence. The overall utility loss for $(x_i, y_k)$ is computed over all sensitive tokens and candidate replacements, ensuring that the semantic structure is preserved as faithfully as possible.

The experiment is performed for multiple values of $\epsilon$, the resulting utility loss stores the variations corresponding to the varying privacy guarantees on the semantic utility. The final matrix provides the insights about the trade-offs between privacy preservation and utility.

---

[2]spaCy: Industrial-Strength Natural Language Processing in Python. Available at: https://spacy.io

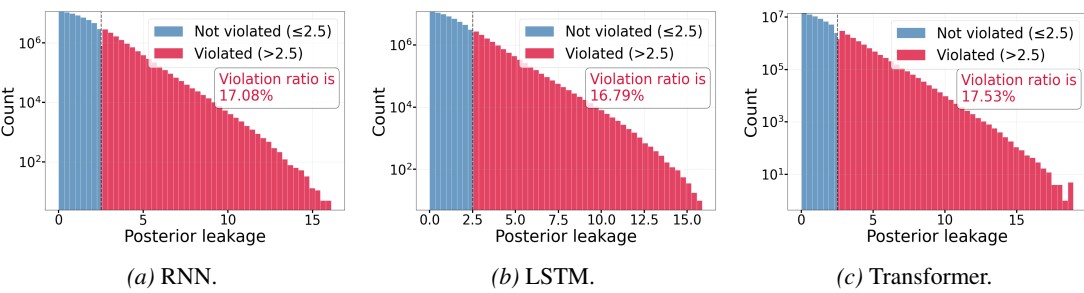

*(a)* RNN.  *(b)* LSTM.  *(c)* Transformer.

*Figure 6.* Examples of mPL distributions derived by different DNN-based inference models.

## F. Additional Experimental Results

### F.1. Examples of mPL Distributions Derived by Different DNN-based Inference Models

Figure 6 provides supplementary results to Figure 4 by visualizing the empirical distribution of mPL values under three learned inference attackers: (a) RNN, (b) LSTM, and (c) Transformer in the AG News dataset. Each figure shows a histogram of per-sample mPL, partitioned into *not violated* versus *violated* regions according to whether mPL exceeds the target budget; the reported violation ratio is the fraction of samples that fall into the violated region. Notably, the violation ratios are non-trivial, underscoring the practical importance of reducing these violations under joint consumption.

### F.2. Recommended $\delta$ Thresholds and Failure Bound Analysis

*Table 3.* Estimated achievable threshold $\tilde{\delta} = 1.05\,\delta^{\star}$ (5% margin).

|  | RNN | | | LSTM | | | Transformer | | |
|---|---|---|---|---|---|---|---|---|---|
| $\epsilon$ | 2.40 | 2.50 | 2.60 | 2.40 | 2.50 | 2.60 | 2.40 | 2.50 | 2.60 |
| AG News | 0.0924 | 0.0876 | 0.0802 | 0.1124 | 0.1032 | 0.0965 | 0.1582 | 0.1566 | 0.1429 |
| IMDB Review | 0.0915 | 0.0859 | 0.0743 | 0.0725 | 0.0649 | 0.0746 | 0.1199 | 0.1029 | 0.1005 |
| Amazon Review | 0.0747 | 0.0689 | 0.0659 | 0.0785 | 0.0667 | 0.0653 | 0.1006 | 0.0895 | 0.0954 |

*Table 4.* Lower bound on $\Pr[p_{\mathcal{X}_\ell^2} \le \delta]$ (calculated by Proposition 3.2). Each entry reports $k$ such that $\Pr[p_{\mathcal{X}_\ell^2} \le \delta] \ge 1 - 10^{-k}$.

|  | RNN | | | LSTM | | | Transformer | | |
|---|---|---|---|---|---|---|---|---|---|
| $\epsilon$ | 2.40 | 2.50 | 2.60 | 2.40 | 2.50 | 2.60 | 2.40 | 2.50 | 2.60 |
| AG News | $4.52 \times 10^5$ | $4.06 \times 10^5$ | $3.41 \times 10^5$ | $6.68 \times 10^5$ | $5.64 \times 10^5$ | $4.93 \times 10^5$ | $1.33 \times 10^6$ | $1.30 \times 10^6$ | $1.08 \times 10^6$ |
| IMDB Review | $3.87 \times 10^5$ | $3.41 \times 10^5$ | $2.55 \times 10^5$ | $2.43 \times 10^5$ | $1.95 \times 10^5$ | $2.57 \times 10^5$ | $6.65 \times 10^5$ | $4.89 \times 10^5$ | $4.67 \times 10^5$ |
| Amazon Review | $3.47 \times 10^5$ | $2.95 \times 10^5$ | $2.71 \times 10^5$ | $3.84 \times 10^5$ | $2.77 \times 10^5$ | $2.65 \times 10^5$ | $6.30 \times 10^5$ | $4.98 \times 10^5$ | $5.67 \times 10^5$ |

For each dataset and privacy budget $\epsilon$, we report the estimated achievable PBmPL threshold $\tilde{\delta} = 1.05\,\delta^{\star}$, where $\delta^{\star}$ is the empirically attainable violation level and the $5\%$ factor provides a small safety margin. Columns correspond to the adversary used for auditing (RNN/LSTM/Transformer). The values are the resulting thresholds (in $[0, 1]$), i.e., the target upper bounds on the PBmPL violation probability under the specified threat model; smaller values indicate stricter privacy targets.

Using Proposition 3.2, in Table 4, we translate each threshold $\delta$ into a concentration-based lower bound on $\Pr\left[p_{\mathcal{X}_\ell^2} \le \delta\right]$, i.e., the probability that the true PBmPL violation probability does not exceed the target. Because the resulting bounds are extremely close to $1$ at our sample sizes, each entry is reported in exponent form: we give $k$ such that $\Pr\left[p_{\mathcal{X}_\ell^2} \le \delta\right] \ge 1 - 10^{-k}$. Larger $k$ therefore indicates higher confidence that the true PBmPL violation probability lies below $\delta$. We set $S_\ell$ to the number of audit samples: AG-News $= 74{,}434{,}304$, IMDB $= 65{,}015{,}552$, Amazon $= 87{,}515{,}904$. The results show that even under these tighter settings, PBmPL failure probabilities remain astronomically small, confirming that violations are virtually impossible at scale.

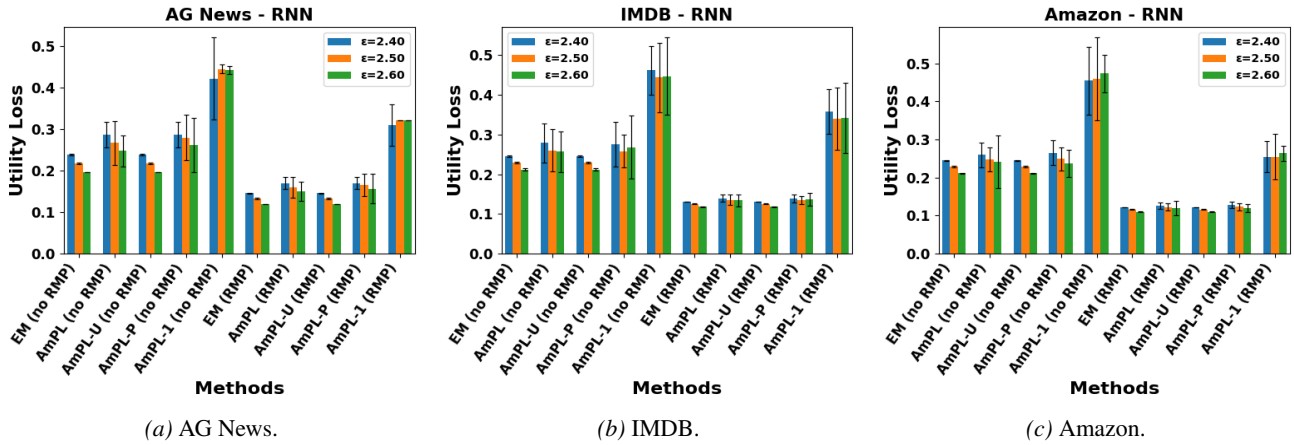

*(a)* AG News.  *(b)* IMDB.  *(c)* Amazon.

*Figure 7.* Expected utility loss (applying RNN as the adversarial model).

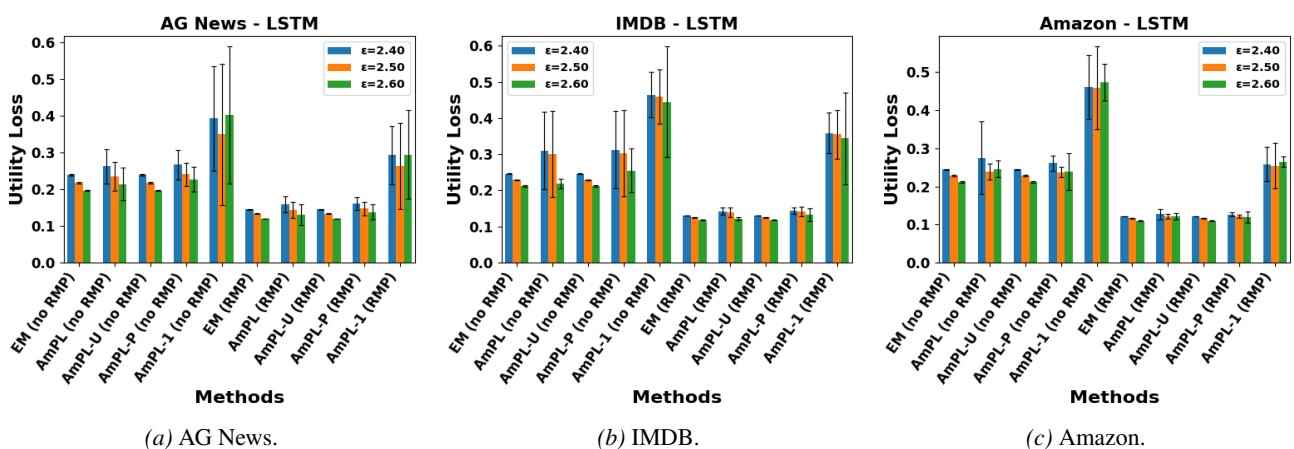

*(a)* AG News.  *(b)* IMDB.  *(c)* Amazon.

*Figure 8.* Expected utility loss (applying LSTM as the adversarial model).

### F.3. Utility Loss

Figures 7–9 report *utility loss* for the same method set and ($\epsilon \in \{2.40, 2.50, 2.60\}$) configuration, differing only by the adversary: Fig. 7 uses a *RNN*, Fig. 8 uses a *LSTM*, and Fig. 9 uses a *Transformer*; in each figure, panels (a)–(c) correspond to AG News, IMDB, and Amazon, and bars are grouped by methods (EM/AmPL variants, with and without the remapping step, RMP). Across all datasets and adversaries, utility loss *monotonically decreases* as $\epsilon$ increases (privacy–utility trade-off); within the AmPL family, AMPL-U consistently achieves the *lowest* or near-lowest loss (by design), while AMPL closely follows with a small gap and AMPL-P typically lie between AMPL and EM. AMPL-1 has the worst utility. Enabling RMP further *reduces* loss compared to the corresponding "no RMP" variants. The *relative ordering* of methods is consistent with the results depicted in Fig. 5 (Transformer), indicating robustness to the attacker model; dataset-wise, Amazon shows the largest absolute losses, followed by AG News and IMDB, but the method ranking and $\epsilon$-sensitivity remain consistent.

We also observe that, across all datasets, adversaries, and $\epsilon \in \{2.40, 2.50, 2.60\}$, enabling RMP yields a large and consistent reduction in utility loss. For the core methods (EM/AmPL/AmPL-U/AmPL-P), the no-RMP losses typically fall in the $\approx 0.22$–$0.30$ range, while the corresponding RMP variants concentrate around $\approx 0.11$–$0.17$. Concretely, on AG News, RMP brings these methods from roughly $\approx 0.24$–$0.29$ (no-RMP) down to $\approx 0.13$–$0.17$ (RMP), i.e., about a *35–45%* reduction; on IMDB, from $\approx 0.23$–$0.29$ down to $\approx 0.11$–$0.14$ (about *45–55%*); and on Amazon, from $\approx 0.22$–$0.27$ down to $\approx 0.10$–$0.13$ (about *45–55%*). For *AmPL-1*, the absolute loss remains higher, but RMP still provides noticeable improvements: from $\approx 0.33$–$0.44$ to $\approx 0.22$–$0.33$ on AG News (*20–35%*), from $\approx 0.40$–$0.47$ to $\approx 0.33$–$0.36$ on IMDB (*20–25%*), and from $\approx 0.45$–$0.49$ to $\approx 0.25$–$0.30$ on Amazon (*35–45%*).

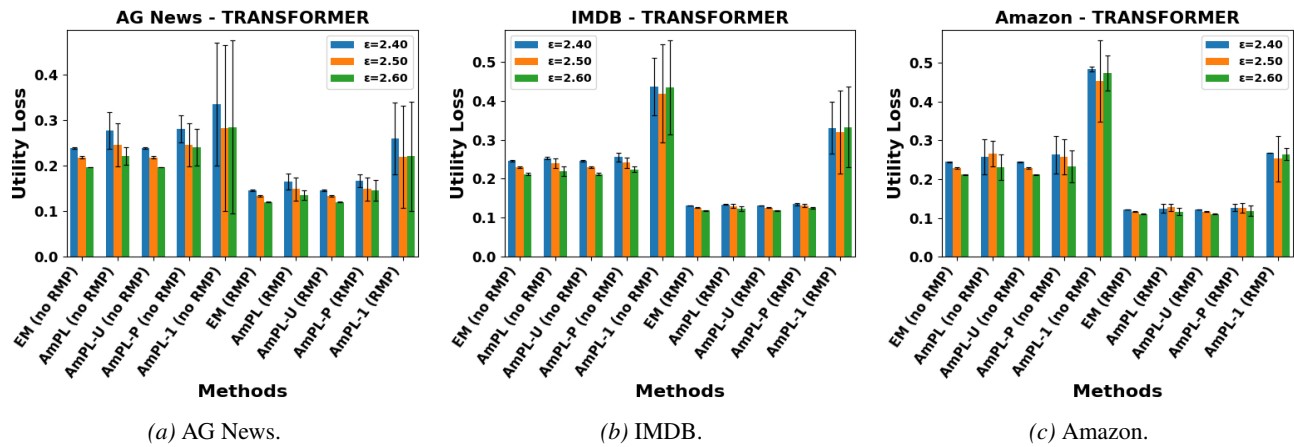

*(a)* AG News.      *(b)* IMDB.      *(c)* Amazon.

*Figure 9.* Expected utility loss (applying Transformer as the adversarial model).

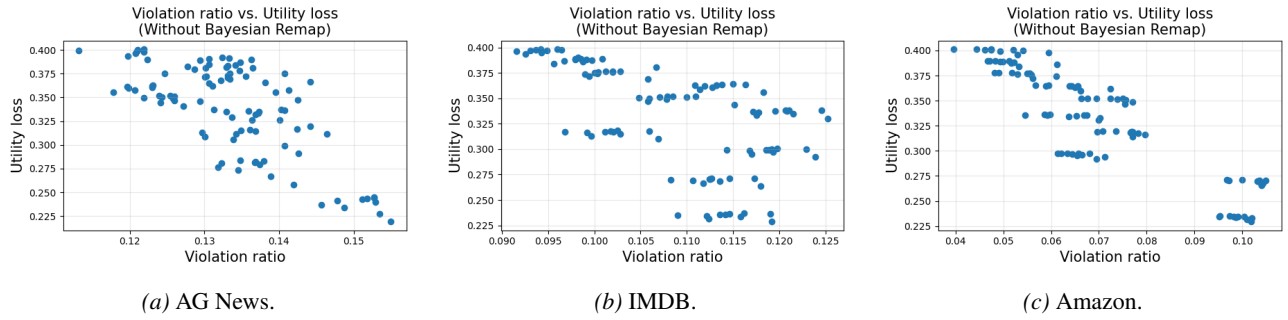

*(a)* AG News.      *(b)* IMDB.      *(c)* Amazon.

*Figure 10.* Trade-off between empirical mPL violation ratio and utility loss for AmPL without Bayesian remap (applying Transformer as the adversarial model).

## F.4. Tradeoff Between Utility and Violation Rate

Figure 10 illustrates the trade-off between utility loss and the empirical mPL violation ratio for AmPL without Bayesian remap. Each point corresponds to one configuration of the mechanism, obtained by varying $\alpha_1, \alpha_2 \in [0.1, 1.0]$ with step 0.1, evaluated at base privacy level $\varepsilon = 2.5$ on 74,434,304 $(x_i, x_j, y)$ triples. The scatter plot shows that as the violation ratio decreases (moving left), the utility loss generally increases, illustrating the expected privacy–utility trade-off: configurations that inject less noise achieve lower utility loss but suffer higher violation ratios, whereas configurations enforcing lower violation ratios incur slightly higher utility loss.

## F.5. Posterior Leakage Comparison with Wider Privacy-Budget Range.

Figure 11 reports the *average* mPL given different privacy budget $\epsilon$ under the RNN attacker, for AG News, IMDB Reviews, and Amazon Reviews. Across datasets, the curve exhibits a clear regime change: mPL stays relatively low and stable for smaller-to-moderate $\epsilon$ (when $\epsilon \leq 2.0$), then rises sharply in a transition region (when $2.0 \leq \epsilon \leq 3.1$), and finally saturates at a much higher level for larger $\epsilon$ (when $\epsilon \geq 3.1$). Figure 12 plots the mPL *violation ratio* versus $\epsilon$ (again under the RNN attacker) for the same three datasets. Viewed together with Figure 11, Figure 12 quantifies how much probability mass lies above the budget $\epsilon$ at each operating point. In particular, the violation ratio is largest around the similar transition region ($2.0 \leq \epsilon \leq 3.1$) suggested by the "leakage cliff" in Figure 11, reflecting that this is where the leakage distribution moves upward fastest relative to the budget. For larger $\epsilon$, the violation ratio decreases, consistent with the fact that the threshold $\epsilon$ itself becomes more permissive even though the average leakage has already entered a high-leakage regime.

Figures 11–12 motivate our choice of $\epsilon \in \{2.4, 2.5, 2.6\}$: Figure 11 reveals a "leakage cliff" where average posterior leakage rises sharply, and Figure 12 shows a corresponding surge in violation ratio in the same transition region. We thus select $\{2.4, 2.5, 2.6\}$ to stay on the low-leakage/low-violation side of this cliff while retaining useful utility; beyond the peak, the violation ratio changes more gradually with $\epsilon$, indicating reduced sensitivity to further noise changes.

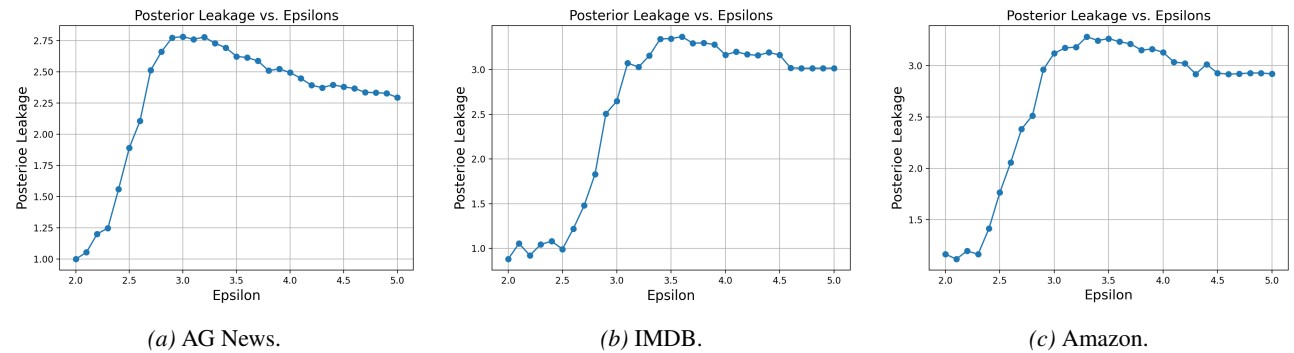

*(a)* AG News.           *(b)* IMDB.           *(c)* Amazon.

*Figure 11.* Average mPL given different $\epsilon$ (applying RNN as the adversarial model).

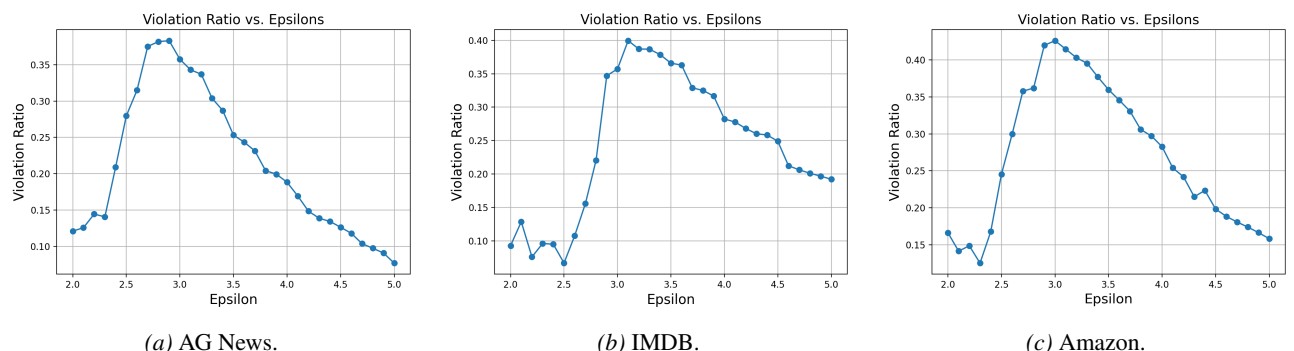

*(a)* AG News.           *(b)* IMDB.           *(c)* Amazon.

*Figure 12.* mPL violation ratio given different $\epsilon$ (applying RNN as the adversarial model).

## F.6. Effect of Attacker Training Data Size

To assess how many supervised pairs the learned adversary requires, we run a learning-curve experiment on AG NEWS. We subsample the adversary's training set to fractions $r \in (0, 1]$ of the original size, retrain the attacker for each $r$, and report (i) attack accuracy, measured by the average cosine similarity between reconstructed and ground-truth embeddings, and (ii) the fraction of $(x_i, x_j, y)$ triples that violate the mPL threshold. Figure 13 shows that performance saturates quickly: using only $\approx 60\%$ of the supervised training pairs already achieves nearly the same cosine accuracy and mPL violation ratio as training on the full dataset.

We also observe that the estimated violation ratio can *slightly increase* with more training data. Since mPL is defined via the deviation of posterior odds $\frac{q_\theta(x_i|y)}{q_\theta(x_j|y)}$ from prior odds $\frac{p(x_i)}{p(x_j)}$, better-trained attackers can extract more information from the perturbed releases and produce sharper posteriors, which may expose additional violations.

## F.7. Attacker Knowledge of the Embedding Model.

In our experiment in Section 4, we adopt a strong, white-box attacker that knows the victim's embedding model; this yields conservative (adversary-favorable) estimates of joint leakage. This assumption is realistic when the encoder is public (e.g., open models or advertised API backends), and even when it is not, an attacker can often train or reuse a surrogate encoder with transferable representations.

Notably, our auditing framework does not require the attacker to know the defender's exact embedding model. If the attacker instead uses a mismatched encoder, inference is typically weaker, yielding smaller posterior leakage and fewer mPL/PBmPL violations; thus, results under a matched (white-box) encoder can be interpreted as conservative upper bounds for less-informed adversaries. Table 5 quantifies this effect by comparing matched versus mismatched embeddings across RNN/LSTM/Transformer attackers (mean $\pm$ std) and metric settings. Overall, mismatched embeddings reduce posterior leakage, with the largest drop observed for RNN/LSTM, while Transformer results are closer under match/mismatch.

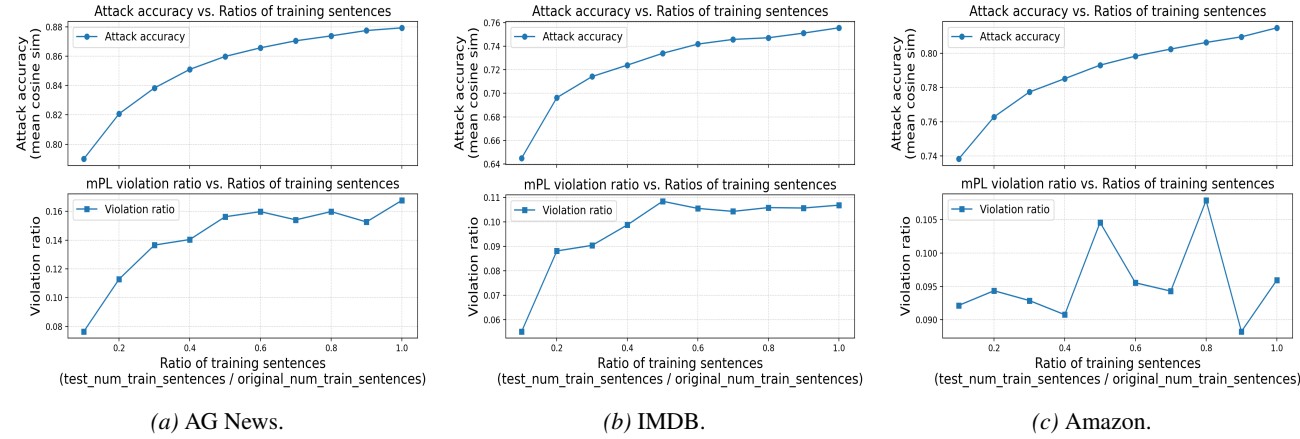

*(a)* AG News.        *(b)* IMDB.        *(c)* Amazon.

*Figure 13.* **Effect of attacker training data size.** Top: attack accuracy (cosine similarity) as a function of the normalized number of training sentences. Bottom: mPL violation ratio as a function of the normalized number of training sentences.

*Table 5.* Posterior leakage under matched vs. mismatched embedding models (mean $\pm$ std).

| **RNN** | $\epsilon = 2.40$ | $\epsilon = 2.50$ | $\epsilon = 2.60$ |
|---|---|---|---|
| matched | $1.2848 \pm 0.0725$ | $1.4344 \pm 0.0362$ | $1.6498 \pm 0.0698$ |
| mismatched | $1.3656 \pm 0.0553$ | $1.4085 \pm 0.0604$ | $1.4602 \pm 0.1237$ |
| **LSTM** | $\epsilon = 2.40$ | $\epsilon = 2.50$ | $\epsilon = 2.60$ |
| matched | $1.2766 \pm 0.0866$ | $1.4079 \pm 0.0829$ | $1.6092 \pm 0.0808$ |
| mismatched | $1.0473 \pm 0.1381$ | $1.1160 \pm 0.0851$ | $1.0761 \pm 0.1669$ |
| **Transformer** | $\epsilon = 2.40$ | $\epsilon = 2.50$ | $\epsilon = 2.60$ |
| matched | $1.4110 \pm 0.0535$ | $1.4646 \pm 0.0631$ | $1.5826 \pm 0.0736$ |
| mismatched | $1.2272 \pm 0.0522$ | $1.2620 \pm 0.0545$ | $1.3028 \pm 0.0596$ |

## F.8. Generality Beyond Text.

Although our experiments focus on textual embeddings, both the joint-leakage notion (mPL) and the AmPL repair framework are modality-agnostic by construction. mPL assumes only (i) a metric $d$ over the secret space and (ii) a task-specific utility loss; AmPL additionally requires (iii) a representation space in which the mechanism operates and (iv) a learned attacker that estimates posteriors from perturbed outputs. None of these components are specific to text. Similar joint-leakage risks arise whenever multiple correlated releases about the same underlying secret are produced in other modalities, for example, multiple views of the same image (vision), longitudinal records for the same entity (tabular or time-series), or multiple recordings of the same speaker/event (audio). In such settings, an adversary can train a multi-input model to aggregate correlated observations and potentially violate per-release privacy guarantees.

To further demonstrate generality beyond text, we evaluate mPL violation ratios on a tabular dataset: the Breast Cancer Wisconsin (Diagnostic) dataset (569 records, 30 continuous features). We z-score standardize features prior to perturbation and treat each record as a single-token example, using the 30-dimensional feature vector as the "token embedding." Table 6 compares PBmPL violation ratios for the EM baseline and AmPL under RNN/LSTM/Transformer attackers across privacy budgets. Across models and budgets, AmPL consistently reduces posterior-leakage violations relative to EM, with an average reduction of 58.1%.

*Table 6.* Breast Cancer Wisconsin (Diagnostic): posterior-leakage violation ratio (%) for EM and AmPL under different learned attackers (mean $\pm$ std).

| **RNN** | $\epsilon = 0.10$ | $\epsilon = 0.20$ | $\epsilon = 0.30$ |
|---|---|---|---|
| EM (mDP) | $35.40 \pm 23.42$ | $19.56 \pm 20.59$ | $14.23 \pm 12.45$ |
| AmPL | $13.42 \pm 19.44$ | $3.66 \pm 4.34$ | $0.98 \pm 2.44$ |
| **LSTM** | $\epsilon = 0.10$ | $\epsilon = 0.20$ | $\epsilon = 0.30$ |
| EM (mDP) | $14.06 \pm 27.64$ | $14.18 \pm 23.37$ | $12.67 \pm 13.70$ |
| AmPL | $4.74 \pm 4.37$ | $3.95 \pm 6.27$ | $2.71 \pm 3.58$ |
| **Transformer** | $\epsilon = 0.10$ | $\epsilon = 0.20$ | $\epsilon = 0.30$ |
| EM (mDP) | $48.49 \pm 23.78$ | $29.88 \pm 25.96$ | $19.49 \pm 12.88$ |
| AmPL | $45.25 \pm 18.27$ | $27.22 \pm 29.23$ | $9.03 \pm 14.15$ |

