# OpenReview forum: "Metric-Normalized Posterior Leakage (mPL): Attacker-Aligned Privacy for Joint Consumption"
_ICML.cc/2026/Conference — Submitted to ICML 2026_

### Official Review · Reviewer_niEK · 2026-03-09

**Soundness:** 3
**Presentation:** 3
**Significance:** 3
**Originality:** 3
**Overall Recommendation:** 4
**Confidence:** 4

**Summary:**

This paper investigates metric differential privacy (MDP), which is an extension of local differential privacy, in the context of privatizing text embeddings. They show (both empirically and theoretically) that standard approaches based on applying an MDP mechanism to each record independently can miss correlations and can lead to apparent violations of the DP guarantee (e.g. the likelihood ratio can be larger than $\varepsilon$). To fix this problem, authors introduce metric-normalized posterior leakage (mPL), which is just the difference in log likelihood ratios before and after seeing the output of the mechanism. This simplifies to MDP under the independent record assumption, but can be large even if MDP is small when correlations are present. Authors establish that mPL is closed under post processing.

However, proving that a mechanism satisfies mPL appears to be difficult, and therefore, instead of providing a mechanism that is provably mPL, authors instead provide a framework for auditing an algorithm that satisfies MDP (note: not mPL!) for its empirical mPL. Authors do this by training an adversarial neural network to reconstruct mechanism inputs from outputs. The empirical mPL is then estimated from this network.

Authors then go further, and demonstrate that this auditing technique can be used in a loop that trades off utility and privacy by carefully changing the mechanism's privacy parameters (for example, $\varepsilon$ in the exponential mechanism). They call this Adaptive mPL (AmPL). The input to this is a mechanism that satisfies MDP and a target mPL guarantee. Inside the loop, the loop carefully tunes the privacy parameters so that the empirical mPL is equal to the target mPL and maximizes utility.

**Compliance With Llm Reviewing Policy:**

Affirmed.

**Final Justification:**

This paper introduces a challenging privacy setting of metric-normalized posterior leakage (mPL), and does not contain theory regarding how to prove that a given mechanism satisfies mPL. The definition of mPL is reasonably straightforward (difference in log likelihoods). The main novelty of this paper, in my opinion, is the trust-and-verify framework, which simultaneously audits a mechanism for mPL and balances its parameters to trade off privacy and utility.

The initial paper contained two major errors that I spotted, both of which have been corrected. I advocate for weak acceptance given these corrections. I will not give this paper a higher score due to the lack of theory for proving if a mechanism satisfies mPL. I question the usefulness of a privacy definition that does not easily lend itself to theoretical analysis, but I can see why, in the absence of theoretical analysis, having a definition of privacy that is auditable is desirable.

**Key Questions For Authors:**

See the weakness section for my questions.

**Limitations:**

Authors could be more open about the difficulty of proving that a mechanism satisfies mPL. But this is minor compared to my other concerns.

**Strengths And Weaknesses:**

Strengths:
- This paper provides a solid mixture of theory (by introducing the novel metric-normalized posterior leakage framework) and experiment (by providing an audit for this new privacy definition).

Weaknesses:
- the proof for Proposition 3.3 references (Bartlett et al. 2017) for its claim that ```with high probability over the draw of the training sample, the excess population cross-entropy of the trained model over the best-in-class predictor decays at a polynomial rate in n```. This claim is crucial for Proposition 3.3 but (Bartlett et al. 2017) never mentions excess population, cross entropy, or best-in-class predictors. It only shows polynomial decay in n for the misclassification probability of AlexNet/ResNet style models (see, for example, Theorem 1.1). I suggest that authors replace this citation with one that actually backs up their claims, or correct me if I am mistaken.
- Authors show that their definition of mPL is closed under post-processing in Proposition 2.4. However, in their implementation of Adaptive mPL, they introduce probabilistic mPL, which is defined identically to probabilistic DP. If I understand Section 3 correctly, the Bayesian Remapping implicitly assumes that probabilistic mPL is also closed under post-processing. This is not necessarily true, see (Meiser 2018) for a simple argument for why probabilistic DP is not closed under post-processing. Can the authors elaborate on what impact this has on their results?

My two weaknesses are major concerns I have about this paper. I will mark this paper as reject for now, but am very open to improving my score if my concerns are addressed.


References:

Bartlett, P. L., Foster, D. J., & Telgarsky, M. J. (2017). Spectrally-normalized margin bounds for neural networks. Advances in neural information processing systems, 30.

Meiser, S. (2018). Approximate and probabilistic differential privacy definitions. Cryptology ePrint Archive. https://eprint.iacr.org/2018/277.pdf

---

> ### Author Rebuttal · Authors · 2026-03-31
>
> Thank you for providing constructive comments/suggestions. Below, we provide the response to the questions and comments.
>
> **Question 1 (Weakness 1): Citation Mismatch in Proposition 3.3**
>
> **Response:** We thank the reviewer for identifying the citation issue. We agree that Bartlett et al. (2017) does not directly support the convergence claim used in our original proof, and we will revise Proposition 3.3 accordingly.
>
> Meanwhile, we clarify that a specific CE-loss learning bound is not central to our contribution. Our focus is not to establish generalization guarantees for a particular model class, but to show how posterior estimation error with a reasonable (e.g., polynomial) convergence rate propagates to mPL. Our analysis already provides this propagation result. To make this distinction explicit and improve the theoretical clarity, we will revise Proposition 3.3 by introducing a third assumption on posterior estimator:
> A3: $E_Y$$[\mathrm{KL}(p(\cdot|Y) || q_{\theta}(\cdot|Y))] \leq O(n^{-\alpha}).$
>
> Under this assumption, the remainder of our proof is unchanged and we will derive the identical conclusion: $\forall \mathbf{x}_i, \mathbf{x}_j$ exists a contant
>
> $C=\frac{2\sqrt{2}}{\gamma d_{min}} > 0$
>
> such that
>
> $E_Y [| mPL(\textbf{x}_i, \textbf{x}_j;Y) - \hat{mPL(\textbf{x}_i, \textbf{x}_j;Y)} |  ] \leq C n^{-\frac{\alpha}{2}}$. This modification isolates our core contribution which is a stability result showing how posterior estimation error propagates to metric normalized posterior leakage.
>
>
> The revised proof sketch and a brief justification of this assumption (A3) are as follows:
> 1. (Use Pinsker to draw a KL bound over the true and estimated posterior) $|p - q| \leq \sqrt{2 KL(p||q)}$
> 2. (Marginalize over Y on both side of 1. Then apply A3) $E_Y[ |p-q| ] \leq \sqrt{2 E[KL(p||q)]} \leq O(n^{-\alpha/2})$
> 3. (Use posterior bound in A1) $|\ln p - \ln q| < \frac{1}{\gamma} |p-q|$
> 4. (Bound the difference between log ratios) $|\ln(\frac{p_i}{p_j}) - \ln(\frac{q_i}{q_j})| \leq \frac{2}{\gamma} |p-q|$
> 5. (Apply metric bound in A2 and definition of mPL) $|mPL - \hat{mPL}| \leq \frac{2}{d_{min} \gamma} |p-q|$
> 6. (Marginalize both side of 5 wrt $Y$ and substitute 2.) $E_Y [mPL - \hat{mPL}] \leq \frac{2 \sqrt{2}}{\gamma d_{min}} O(n^{\frac{-\alpha}{2}}).$
>
> To justify A3, note that Lemma D.1shows that excess CE is exactly the expected KL divergence. Classical learning theory (e.g., Rademacher-complexity-based bounds; see Chapter 26 of Understanding Machine Learning [1]) exhibits polynomial finite-sample scaling (typically on the order of $O(n^{-0.5})$. We use this only as motivation and do not rely on it to derive KL convergence. Instead, A3 adopts a polynomial form as a baseline and parameterizes the theorem, allowing any posterior convergence rate to be plugged into our analysis.
>
> [1] S. Shalev-Shwartz and S. Ben-David, Understanding Machine Learning, 2014.
>
> **Question 2 (Weakness 2): Probabilistic mPL may not be closed under post-processing**
>
> **Response:** We thank the reviewer for this important observation. We agree that Proposition 2.4 establishes post-processing invariance only for bounded mPL, and does not imply automatic preservation of PBmPL under Bayesian remapping. We will revise the paper to remove any such implication and make this distinction explicit.
>
> For the updated experiments, we directly *re-audited the remapped mechanism* and found that, under all evaluated attacker families and privacy budgets, the observed PBmPL violation ratio is consistently *lower after remapping*. Averaged across $\epsilon = 2.4, 2.5, 2.6$, the PBmPL violation ratio decreases from *8.26\% to 3.24\%* for RNN, from *9.91\% to 5.87\%* for LSTM, and from *14.53\% to 8.73\%* for Transformer after remapping.
>
> We will revise the paper so that any privacy claim after remapping is supported by a fresh empirical audit of the remapped mechanism, rather than inherited from Proposition 2.4. Accordingly, we do not claim general preservation of PBmPL under remapping.
>
> **Illustrative results (violation ratio before → after remapping):**
>
> AGnews
> |Attacker|ε=2.40|ε=2.50|ε=2.60|
> |-|-:|-:|-:|
> |RNN|8.80±1.30→3.02±1.48|8.34±2.02→3.28±1.78|7.64±0.53→3.42±0.70|
> |LSTM|10.70±2.66→6.08±1.52|9.83±2.29→5.85±1.27|9.19±0.95→5.67±0.85|
> |Transformer|15.07±1.10→8.81±1.00|14.91±2.62→9.09±2.35|13.61±2.33→8.28±1.66|
>
> IMDB
> |Attacker|ε=2.40|ε=2.50|ε=2.60|
> |-|-:|-:|-:|
> |RNN|8.71±2.90→2.87±1.92|8.18±1.46→2.62±0.87|7.08±1.14→2.23±1.38|
> |LSTM|6.90±3.01→3.64±1.99|6.18±3.47→3.14±3.12|7.10±2.89→3.20±1.39|
> |Transformer|11.42±0.98→5.42±1.28|9.80±0.91→4.19±1.28|9.57±0.36→4.26±1.09|
>
> Amazon Review
> |Attacker|ε=2.40|ε=2.50|ε=2.60|
> |-|-:|-:|-:|
> |RNN|7.11±2.21→2.42±1.00|6.56±1.13→2.28±0.67| 6.28±2.75→2.47±1.29|
> |LSTM|7.48±2.85→4.70±0.73|6.35±1.77→3.56±1.02|6.22±2.07→3.33±0.94 |
> |Transformer|9.58±4.62→4.67±2.51|8.52±2.90→4.08±2.48|9.09±2.19→4.03±1.69|

---

> > ### Author Rebuttal · Reviewer_niEK · 2026-04-02
> >
> > Thank you for addressing my concerns. I have raised my score from 2: Reject to 4: Weak Accept.

---

> > > ### Author Response · Authors · 2026-04-02
> > >
> > > **Response:** Thank you for the thoughtful follow-up and for increasing your score. We truly appreciate your careful consideration of our rebuttal and your positive evaluation of the paper. We are also very grateful for your constructive feedback throughout the review process.

---

### Official Review · Reviewer_UpnJ · 2026-03-11

**Soundness:** 3
**Presentation:** 3
**Significance:** 2
**Originality:** 3
**Overall Recommendation:** 4
**Confidence:** 4

**Summary:**

Most of differential privacy assumes that records/datapoints are sampled iid. User-level differential privacy accounts for dependency between the data of a user, but still assumes user-level data are iid. This paper studies privacy guarantees when there are correlations between datapoints that an attacker can use to extract training data. Authors state their approach is most useful in cases where we lack user identifiers and thus cannot resort to user-level differential privacy.

The authors propose a defintion of privacy that considers the joint leakage of information due to data correlations. Their definiton is a generalization of metric-level DP, useful in cases where users privatize their data locally before sending them for aggregation to the server. It is hard to obtain a privacy-preserving mechanism which satisfies the theoretical definition thus the authors introduce an iterative adversary-in-the loop approach that iteratively find the right level of gaussian noise to add to data so as to bound the empirical probability of violating the definition from an adversary. The adversary is instantiated as a neural network for reconstructing original data from noisy data (using correlations between datapoints) with more complex models representing stronger adversaries.

They evaluate their methodology on 3 datasets and show that a mechanism which only satisfies metric-DP (and assumes iid data) leaks more information that their methodology which accounts for data correlations.

**Compliance With Llm Reviewing Policy:**

Affirmed.

**Final Justification:**

Thank you for this response which answered all of my concerns.

Thank you for pointing to Appendix C.2 and for re-clarifying the use cases where joint-mDP would be useful. I agree that the cases where several accounts are owned by the same individual or when an account mentions sensitive information about an individual other than the owner are not covered by existing differential privacy frameworks. Since achieving theoretical guarantees of DP in these settings is hard, I find that the empirical methods offered by the authors are well-motivated. I have increase my score from 2-> 4 as a result.

**Key Questions For Authors:**

I welcome authors' comments on use cases where their algorithm would be relevant.

In addition, why is the choise of epsilon in the case study limited to the range of [2.4, 2.6]? Wouldn't it be more interesting to consider a wider range of epsilon?

Finally, can you please provide non-private baselines for the utility of the trained classifiers to contextualize the private baselines?

**Strengths And Weaknesses:**

Strengths
- This paper tackles a challenging and less studied issue in the differential privacy literature, which is the issue of privacy leakage due to data correlations.
- The privacy defintions introduced are sound. Given the difficulty of operationalizing such definitions, the authors provide a sensible algorithm for empirically and approximately satisfying the definition.

Weaknesses
- I am not as convinced about the types of use cases where the algorithm is needed. E.g, the work assumes a local privacy model, where users randomize their data before sending them to the server, however in such cases the server should have user identifiers if its communicating with users, and thus user-level differential privacy can be applied. The authors do not compare their method to user-level differential privacy. Moreover, "organizations with incentives of collecting and analyzing user data" will almost certainly have access to user identifiers. A case where we lack user identifiers is with internet-scraped data for training LLMs, however, in this case a central rather than local model of privacy is more relevant. The authors mention that their method can be applied alongside user-level differential privacy, but there is no evaluation of whether user-level differential privacy is enough to prevent the leakage due to data correlations and thus whether any additional measures are needed.

- There is not much discussion of the communication cost between users and servers until convergence to a desired level of privacy noise is reached. There is also not much discussion on the cost of retraining the adversarial model. These considerations would make the algorithm very hard to apply in practice.

---

> ### Author Rebuttal · Authors · 2026-03-31
>
> Thank you for providing constructive comments/suggestions. Below, we provide the response to the questions and comments.
>
> **Question 1 (Weakness 1): Why user-agnostic joint leakage requires more than user-level DP**
>
> **Response:** We thank the reviewer for this important point. We agree that *when reliable user identifiers and trusted grouping assumptions are available, user-level DP is a natural mitigation*. Our goal, however, is to address a different setting: *user-agnostic joint leakage*, where the defender cannot reliably determine how records should be grouped, while the adversary may still aggregate correlated releases. As discussed in **Appendix C.2**, this setting arises when a single account may mention multiple individuals or secrets (without reliable identifiers), and when linking related records is itself part of the adversarial inference task.
>
> This is also why we did not include a user-level DP baseline in the current experiments. The public datasets used in our case study do *not* provide reliable user identifiers or ground-truth group labels, so constructing such a baseline would require additional task-specific grouping heuristics that are outside the threat model studied here. Our intended use cases are therefore settings with *local / record-level releases but ambiguous grouping*, such as cross-account text sharing, cross-document mention protection, and *PII/PoII perturbation before downstream NLP analytics*. This use case is consistent with prior text-privacy work on *client-side / on-device privatization before provider-side analysis*, including on-device text privatization where sensitive text never leaves the device and the provider only receives privatized text to “train models on and analyze” (Carvalho et al., 2021), as well as user-side text sanitization for downstream NLP pipelines (Yue et al., 2021). More broadly, our framework targets *local, mDP-native auditing under ambiguous grouping* and supports learned attacker models when exact joint inference is intractable or the true dependence structure is unknown. Prior local/privacy-preserving NLP work has also considered user-side text sanitization before downstream analysis, while generalized/metric-DP text work has been motivated by sanitizing *patient health records* and *government documents* (Fernandes et al., 2019).
>
> We therefore view *user-level DP as complementary rather than competing* with our framework: when reliable identifiers are available, user-level accounting is appropriate; when grouping is ambiguous, our framework is intended to audit the additional leakage that may arise from *joint consumption of correlated releases*. We will revise the motivation and threat model accordingly.
>
> **Question 2 (Weakness 2): Offline Calibration vs. Online Communication**
>
> **Response:** Thank you for this question. AmPL is intended as an *offline attacker-aware calibration* for a *user-side perturbation mechanism*: users perturb each record locally once before transmission, while a strong attacker is used offline to audit leakage and tune parameters. Thus, deployment-time communication is still *one perturbed release per record*. We will revise the text and figure to clarify this distinction.
>
> Moreover, we agree that adversarial retraining adds *offline computational overhead*, and we will state this more clearly as a limitation. Still, the audit uses random sampling rather than exhaustive evaluation, and **Appendix F.6** shows that training already saturates at about *60%* of the supervised pairs.
>
> **Question 3: Choice of $\epsilon$ range in the case study**
>
> **Response:** We agree that a wider $\epsilon$ range is important, and this is already examined in **Appendix F.5**. There, we evaluate a broader range and show that both the *average mPL* and the *mPL violation ratio* exhibit a clear transition: leakage stays relatively low for smaller $\epsilon$, rises sharply in a transition region, and then saturates for larger $\epsilon$. Our choice of $\epsilon \in \{2.4, 2.5, 2.6\}$ was therefore intentional: these values lie in the most sensitive part of that transition, where leakage changes rapidly with $\epsilon$, making them the most informative operating points for method comparison. We will make this motivation more explicit in the main text.
>
> **Question 4: Non-private baseline**
>
> **Response:** Thank you for this suggestion. We would like to clarify that our utility metric is *not classifier accuracy*, but the *perturbation-induced utility loss* used throughout the paper. Specifically, $c_{x,y}$ measures the loss incurred by releasing $y$ instead of the true input $x$. Under this definition, the *non-private baseline has utility loss 0*, so all reported utility-loss values should be interpreted as degradation *relative to the non-private case*. We agree that this reference point should be stated more explicitly in the revision.

---

> > ### Author Rebuttal · Reviewer_UpnJ · 2026-04-02
> >
> > Thank you for this response which answered all of my concerns.
> >
> > Thank you for pointing to Appendix C.2 and for re-clarifying the use cases where joint-mDP would be useful. I agree that the cases where several accounts are owned by the same individual or when an account mentions sensitive information about an individual other than the owner are not covered by existing differential privacy frameworks. Since achieving theoretical guarantees of DP in these settings is hard, I find that the empirical methods offered by the authors are well-motivated. I have increase my score from 2-> 4 as a result.

---

> > > ### Author Response · Authors · 2026-04-02
> > >
> > > **Response:** Thank you for the thoughtful follow-up and for raising your score. We sincerely appreciate your careful consideration of our rebuttal, as well as your positive assessment of the paper. We are also grateful for your constructive feedback throughout the review process.

---

### Official Review · Reviewer_rdvs · 2026-03-12

**Soundness:** 3
**Presentation:** 3
**Significance:** 3
**Originality:** 3
**Overall Recommendation:** 3
**Confidence:** 4

**Summary:**

This paper introduces a novel metric for quantifying privacy violation within the framework of metric-based differential privacy. Unlike previous measures that rely solely on posterior information, the proposed metric incorporates both prior and posterior knowledge, scaled by distance. Building on this privacy violation measure, the paper presents an improved framework that achieves lower privacy violation under the new metric. For utility evaluation, the study explores the application of this framework to LLM embeddings. Experimental results validate the effectiveness of the proposed approach in mitigating privacy violations.

**Compliance With Llm Reviewing Policy:**

Affirmed.

**Final Justification:**

This is a worthwhile idea for measuring privacy leakage. However, as the authors also acknowledge, they are currently unable to obtain a theoretical result and can only support the claim empirically. It would strengthen the work considerably if they could further develop some theoretical guarantee or analytical result.

**Key Questions For Authors:**

1. A key concern is the reliance on posterior information. It remains unclear how to determine whether a model has been sufficiently trained to be suitable for the privacy violation measure. Without a clear criterion for model adequacy, any measured privacy loss may be unreliable or unconvincing.

2. Since the proposed privacy violation measure stems from the concept of "joint consumption" yet remains inherently data-dependent, the paper does not disentangle the extent to which observed privacy loss is attributable to joint consumption processes versus the underlying prior distribution.

**Limitations:**

The paper does not discuss the limitations. Please answer the questions above.

**Strengths And Weaknesses:**

The paper presents a novel perspective on privacy violation within the context of metric-based differential privacy, arguing that privacy can be compromised through “joint consumption”—the accumulation of observations that enable inference of original items via temporal correlations. The authors provide a theoretically rigorous introduction to both the proposed privacy measure and its corresponding perturbation framework. However, my review raises two primary concerns. First, the posterior is inherently tied to the capability of the model, which may disadvantage under-trained models. This becomes particularly problematic given that, in standard differential privacy, an adversary is typically assumed to have unbounded computational resources. Second, because the proposed privacy measure is data-dependent (i.e., informed by the prior), it remains unclear how much of the privacy violation stems from joint consumption versus the underlying data distribution itself.

---

> ### Author Rebuttal · Authors · 2026-03-31
>
> Thank you for providing constructive comments/suggestions. Below, we provide the response to the questions and comments.
>
> **Question 1 (Weakness 1): Model adequacy in posterior leakage estimation.**
>
> **Response:** We agree that attacker adequacy should be assessed explicitly. Our view is that a learned attacker is adequate when *increasing its strength no longer materially changes the estimated leakage*. We use this criterion because the goal of the audit is not to prove that the learned attacker is globally optimal, but to check whether the privacy estimate is still limited by attacker weakness. Since our audit is explicitly *adversary-dependent*, and stronger attackers may reveal additional violations, a natural practical test is whether the estimated leakage stabilizes as attacker strength increases. Quantifying adequacy *directly* with a single absolute metric is difficult in our setting, because for expressive DNN attackers the exact posterior, and hence the exact mPL, is generally intractable, so there is no gold-standard leakage target to compare against. Moreover, average model-fit metrics alone are not sufficient, since a slightly stronger attacker may still uncover additional *tail violations* relevant to PBmPL even when its average prediction quality changes little.
>
> In the current draft, the appendix already includes two diagnostics in this spirit. First, **Appendix F.6** varies the attacker training data size and shows that both attack accuracy and the mPL violation ratio *saturate quickly*: using about *60% of the supervised training pairs* already achieves nearly the same values as the full dataset. Second, **Appendix F.7** studies weaker attacker knowledge via mismatched encoders; the matched white-box setting used in the main paper is intentionally conservative and generally yields higher leakage.
>
> In the revision, we will make this adequacy criterion more explicit: we will present the audit as based on the *strongest tested attacker suite*, and clarify that attacker adequacy can be diagnosed empirically by checking whether the estimated leakage is stable as attacker data, architecture strength, and attacker knowledge increase.
>
> **Question 2 (Weakness 2): Disentangling joint consumption from prior dependence.**
>
> **Response:** We agree that this distinction should be stated more explicitly. Our intention is that *mPL measures posterior sharpening relative to the prior baseline, not the prior itself*. By Definition 2.2, mPL is the change from prior odds to posterior odds after observing the release. If the observation provides no information, then posterior equals prior and mPL is zero, regardless of any dependence already present in the prior.
>
> What we call *joint-consumption leakage* is therefore the *additional posterior sharpening caused by jointly observing dependent releases*. The role of prior dependence is not to be counted as leakage by itself, but to determine how strongly multiple observations can combine once they are consumed together. This is already reflected in two parts of the paper. First, for a *single observation*, bounded mPL is equivalent to mDP, and under *independent releases*, individual mPL bounds remain sufficient for joint observation. Second, in the correlated toy example, each individual perturbed record satisfies the mPL bound ($0.944 < \epsilon$), while the joint observation raises mPL to $1.846 > \epsilon$. Thus, the excess leakage is not attributed to the prior alone, but to the *additional posterior update enabled by joint observation under dependence*.
>
> We will revise the paper to make this interpretation explicit. In particular, we will clarify that mPL treats the prior as the baseline, and that the paper’s novelty is to quantify the *incremental leakage revealed by joint consumption of dependent releases*.

---

> > ### Author Rebuttal · Reviewer_rdvs · 2026-04-01
> >
> > This is a worthwhile idea for measuring privacy leakage. However, as the authors also acknowledge, they are currently unable to obtain a theoretical result and can only support the claim empirically. It would strengthen the work considerably if they could further develop some theoretical guarantee or analytical result.

---

> > > ### Author Response · Authors · 2026-04-02
> > >
> > > **Response:** We thank the reviewer for this suggestion. We would like to clarify that the paper already contains a theoretical result in **Proposition 3.3** (with proof in **Appendix D.5**), which bounds the error of the estimated mPL in terms of the learning/estimation error of the posterior model. In the revision, we will further sharpen this result by making its assumption explicit.
> > >
> > > Specifically, we will state Proposition 3.3 as a **conditional error-propagation result**: if the posterior estimator satisfies **Assumption (A3)**, $E_Y[\mathrm{KL} (p(\cdot\mid Y)|q_\theta(\cdot\mid Y))] \le O(n^{-\alpha})$,
> > > then there exists a constant $C>0$ such that
> > > $E_Y[|mPL(\mathbf{x}_i,\mathbf{x}_j;Y)-\widetilde{mPL}(\mathbf{x}_i,\mathbf{x}_j;Y)|] \le Cn^{-\alpha/2}$.
> > >
> > > Here,
> > >
> > > - $p(\cdot\mid Y)$ denotes the true posterior distribution over secrets given observation $Y$
> > > - $q_\theta(\cdot\mid Y)$ denotes the posterior estimated by the learned attacker
> > > - $mPL(\mathbf{x}_i,\mathbf{x}_j;Y)$ is the **true metric-normalized posterior leakage** between secrets $\mathbf{x}_i$ and $\mathbf{x}_j$ under observation $Y$
> > > - and $\widetilde{mPL}(\mathbf{x}_i, \mathbf{x}_j; Y)$ denotes its estimate induced by the estimated posterior.
> > >
> > > Thus, our framework is not supported only empirically: it already includes a theoretical guarantee showing how posterior estimation error induces a controlled error in the estimated mPL.
> > >
> > > In the revision, we will further strengthen this result by making its assumption more explicit and robust.

---

### Official Review · Reviewer_m1kT · 2026-03-13

**Soundness:** 3
**Presentation:** 3
**Significance:** 3
**Originality:** 3
**Overall Recommendation:** 4
**Confidence:** 3

**Summary:**

This paper revisits metric differential privacy (mDP) under joint observations. It argues that per-record mDP may fail to capture privacy risks when observes multiple dependent releases jointly. To address this issue, the authors propose metric-normalized posterior leakage (mPL) to quantify joint posterior leakage. For independent secrets, they show that bounded mPL is equivalent to per-record mDP. For correlated secrets, they introduce probabilistically bounded mPL (PBmPL) as a practical relaxation for privacy auditing, and further propose Adaptive mPL (AmPL) to audit and control privacy risk. Case studies on PII/PoII embedding protection show improved privacy auditing performance.

**Compliance With Llm Reviewing Policy:**

Affirmed.

**Final Justification:**

The paper remains **Weak Accept** from my perspective. It is technically meaningful and reasonably sound, with clear originality in extending privacy auditing from single releases to joint observations under mDP, although its significance is somewhat limited by the evaluation scope and practical applicability concerns noted in my original review. The rebuttal addressed my main concerns well, clarified the paper’s scope and positioning, and reinforced rather than changed my original positive assessment, so I keep my rating at Weak Accept.

**Key Questions For Authors:**

1. Is AmPL intended to guide the design of privacy-preserving mechanisms for users, such as PII perturbation?
2. Why are the experiments limited to the text modality? Is this common practice in related work, or is there a particular reason for this design choice?
3. Is this the first work to quantify privacy leakage under joint observations of dependent releases? If not, how does it differ from prior work? Are those prior approaches also applicable under the mDP framework?

**Limitations:**

yes

**Strengths And Weaknesses:**

## Strengths

1. **One strength of this paper is its soundness.** The motivations of this paper are clear, and the methodology is well-motivated. In particular, Proposition 3.3 is important, as it helps justify the use of empirical evaluations in AmPL.
2. **Another strength of this paper is its originality.** The authors revisit the limitations of mDP under joint observations and extend privacy auditing from single to multiple dependent releases. I think this is a natural and meaningful extension of prior research in this area.

## Weakness
1. **Methodology**. I still find AmPL's privacy leakage quantification somewhat weak, even considering Proposition 3.3. In particular, it relies on empirical estimation with strong learned attackers. The current presentation also makes the auditing and perturbation steps seem to take place on the user side. If this is indeed the intended setting,  the method may implicitly require users to have sufficiently large auxiliary data, which raises concerns about its practical applicability.
2. **Experiments**. The experimental evaluation is somewhat weak, as it is limited to the text modality. Since AmPL does not appear to be modality-dependent, it would be more convincing to extend the evaluation to other modalities. In addition, the paper would benefit from comparisons with stronger or more relevant baselines.
3. **Figure Presentation.** Some figures rely primarily on color to differentiate key elements, which makes them difficult to understand in black-and-white or grayscale printouts. This issue is particularly noticeable in Figs. 1, 3 (PII / PoII), 6, 8, 9. The authors should revise the figures to ensure that important elements can be distinguished without relying solely on color.

---

> ### Author Rebuttal · Authors · 2026-03-30
>
> Thank you for providing constructive comments/suggestions. Below, we provide the response to the questions and comments.
>
> **Question 1 (Weakness 1): AmPL as a user-side perturbation design framework**
>
> **Response:** Yes. *AmPL is intended to help users/system design and calibrate privacy-preserving perturbation mechanisms for sensitive records to release*. We would also like to clarify that our objective is *attacker-aware protection*. Since joint-consumption leakage depends on the inference capability of plausible attackers, AmPL calibrates the mechanism against strong learned attackers rather than assuming attacker ignorance. In the main experiments, we therefore use strong attacker models (*RNN/LSTM/Transformer trained on relatively large datasets*) as *conservative audits*.
>
> To make this point more explicit, the appendix already includes two additional studies, which we will highlight more prominently in the main text in the revised version:
>
> - **Appendix F.6 (reduced attacker training data):** the attacker-data ablation shows that attack performance saturates quickly. Using only about *60% of the supervised training pairs* already achieves nearly the same attack accuracy and mPL violation ratio as using the full dataset.
>
> - **Appendix F.7 (weaker attacker knowledge):** we further test weaker attacker knowledge through *mismatched embedding models*. The matched-encoder setting used in the main paper is intentionally conservative, while the mismatched setting generally yields lower posterior leakage. For example, at $\epsilon=2.6$, the leakage drops from *1.6092 to 1.0761* for LSTM, from *1.5826 to 1.3028* for Transformer, and from *1.6498 to 1.4602* for RNN.
>
> **Question 2 (Weakness 2): Choice of text as the experimental modality**
>
> **Response:** Our main case study uses text embeddings because embeddings are a common interface in deployed NLP systems, and the PII/PoII split provides a natural example of multi-level protection. However, the framework itself is *not limited to text*. mPL is defined only through a secret-space metric and the induced posterior, and therefore applies more broadly whenever these two ingredients are available. We realize this point may be easy to miss in the current draft, so we would like to highlight that **Appendix C.3 and Appendix F.8 already make this modality-agnostic scope explicit** and include a non-text evaluation on the *Breast Cancer Wisconsin* dataset. In that tabular setting, *AmPL reduces posterior-leakage violations relative to EM by an average of 58.1%*, providing concrete evidence that the framework extends beyond text embeddings and remains effective in a different data modality. We will make this broader applicability more visible in the main paper and clarify that text is used in the current submission as a representative case study, rather than as a restriction of the framework.
>
> **Question 3: Applicability of prior correlated-data privacy approaches to mDP**
>
> **Response:** Prior work has indeed studied privacy under correlated data and dependent observations, most notably through central-DP-style frameworks like *Pufferfish (Kifer & Machanavajjhala, 2012)*. Our claim is narrower: *to the best of our knowledge, this is the first mDP-based, attacker-audited framework for quantifying posterior leakage under joint observation of dependent releases*. The key reason central-DP-style approaches are not directly applicable is that they address a DP object and a different release model. Pufferfish is formulated in a curator-side Bayesian framework and it is designed to reason about inference from curator releases under explicitly modeled correlations. By contrast, mDP is a *local, metric-based* privacy notion: the user perturbs their own secret before release, and the privacy guarantee varies with the distance between secrets in the underlying metric space. This metric semantics is fundamental in mDP and is not inherited automatically from central correlated-data privacy definitions.
>
> Our framework is therefore not simply “Pufferfish applied to mDP.” Instead, it is designed to quantify leakage for *user-side perturbed releases* under dependence while preserving the metric structure of mDP, and it is proved to recover mDP in the *single/independent-release* regime. In addition, prior posterior-based approaches usually rely on an *explicit generative model* for the dependence structure, whereas our main instantiation supports *attacker-audited leakage quantification under implicit dependencies* via learned posterior estimators when exact joint Bayesian inference is intractable or the true joint prior is unknown or misspecified.
>
> We will revise the related-work and contribution sections to make these distinctions clearer.
>
> **Weakness 3: Figure revision**
>
> **Response:** We will revise the figures so that key elements can be clearly distinguished without relying solely on color.

---

> > ### Author Rebuttal · Reviewer_m1kT · 2026-04-02
> >
> > The rebuttal clarifies my main questions satisfactorily. While it does not substantially change my overall assessment, it reinforces my original positive view of the paper. I therefore keep my rating unchanged.

---

> > > ### Author Response · Authors · 2026-04-02
> > >
> > > **Response:** Thank you for the thoughtful follow-up. We appreciate your careful reading of the rebuttal. We also appreciate your positive assessment of the paper and your constructive feedback throughout the review process.

---

### Decision · Program_Chairs · 2026-04-30

**Decision:**

Reject

**Comment:**

The authors study the important problem of metric Differential privacy which is a generalization of DP involving a metric where privacy loss scales with the distance between two adjacent samples. They present a relaxation of mDP based on an analysis of the posterior probability inferred by an adversary observing the output of the algorithm. The method is based on empirically auditing the output distribution using an attacker algorithm (trained on the data). The authors introduce several relaxations to the theoretical guarantees to make the computation feasible (including not bounding the privacy loss for all pairs of adjacent datasets but for samples draw from a distribution). The reviewers appreciated the area and found the paper interesting. The empirical results were of interest to most reviewers. However, the reviewers also identified some limitations in the work. The main limitation that many reviewers identified is the lack of strong theoretical privacy guarantees. The guarantees are based on an empirical estimation of probabilities through a model so the "epsilon" loss in the DP relaxation is not guaranteed and can be affected by sampling, choices of the adversary and other empirical factors. The author present theoretical asymptotic results that under certain circumstances the sampling eventually converges to the right estimation but these results do not provide concrete guarantees for one specific instantiation. Moreover, a reviewer identified some aspects of the theoretical statements that required correction (as discussed in the rebuttal). Finally, the practical applicability of the auditing framework was also found not convincing by some reviewers.